# Conformations of Bcs1L undergoing ATP hydrolysis suggest a concerted translocation mechanism for folded iron-sulfur protein substrate

Jingyu Zhan [1], Allison Zeher[1,2], Rick Huang[1,2], Wai Kwan Tang [1], Lisa M. Jenkins [1] & Di Xia [1] ✉

The human AAA-ATPase Bcs1L translocates the fully assembled Rieske iron-sulfur protein (ISP) precursor across the mitochondrial inner membrane, enabling respiratory Complex III assembly. Exactly how the folded substrate is bound to and released from Bcs1L has been unclear, and there has been ongoing debate as to whether subunits of Bcs1L act in sequence or in unison hydrolyzing ATP when moving the protein cargo. Here, we captured Bcs1L conformations by cryo-EM during active ATP hydrolysis in the presence or absence of ISP substrate. In contrast to the threading mechanism widely employed by AAA proteins in substrate translocation, subunits of Bcs1L alternate uniformly between ATP and ADP conformations without detectable intermediates that have different, co-existing nucleotide states, indicating that the subunits act in concert. We further show that the ISP can be trapped by Bcs1 when its subunits are all in the ADP-bound state, which we propose to be released in the apo form.

AAA (ATPases Associated with various cellular Activities) proteins are molecular machines that are ubiquitously present in all kingdoms of life, performing essential functions in diverse cellular pathways from protein homeostasis, membrane fusion, and DNA replication to substrate translocation[1,2]. They are homo-oligomeric, frequently hexamers, with each subunit consisting of an N-terminal functional domain followed by one or two AAA ATPase motor domains. The involvement of AAA proteins in various cellular pathways is related to their ability to interact with a plethora of protein or DNA substrates, acting as translocases and helicases[1,2]. Cryo-electron microscopy (cryo-EM) analyses of various AAA proteins in the presence of substrates have revealed that AAA proteins engage substrates ubiquitously by a threading mechanism, in which subunits of AAA proteins assemble in a spiral staircase formation, with each subunit captured interacting with an amino acid residue of an unfolding protein polypeptide or with a DNA nucleotide and progressing in a hand-over-hand fashion. However, the debate over whether AAA proteins carry out ATP hydrolysis via a sequential[3-15] or a probabilistic mechanism remains[16-21]. It is important to note that subunits of various nucleotide states (apo, ADP, and ATP) coexist in the same complex.

However, the widely adopted threading model of the AAA protein function seems inconsistent with the proposed function of Bcs1, a yeast AAA protein that translocates fully folded Rieske iron-sulfur protein (ISP or Rip1 in yeast) across the mitochondrial inner membrane (MIM)[22,23]. The ISP is an essential subunit of mitochondrial Complex III, also known as the cytochrome $bc_1$ complex (cyt $bc_1$) or ubiquinol-cytochrome c oxidoreductase, which catalyzes the reaction of reducing cyt c by ubiquinol and couples this reaction to proton pumping across the MIM, contributing to the membrane potential for various cellular activities including ATP synthesis[24]. In metazoan organisms, the ISP subunit is incorporated into the core assembly of cyt $bc_1$, as a precursor protein, at a late stage of the complex's assembly

[1]Laboratory of Cell Biology, Center for Cancer Research, National Cancer Institute, National Institutes of Health, Bethesda, MD 20892, USA. [2]NIH Intramural Cryo-EM Consortium (NICE), Bethesda, MD, USA. ✉e-mail: xiad@mail.nih.gov

(Supplementary Fig. 1A), which is subsequently processed into the mature ISP (residues 79–274 of the precursor, Supplementary Fig. 1B) and subunit 9 (residue 1–78)[25]. The mature ISP has 196 residues and consists of three domains (Supplementary Fig. 1B, C): a C-terminal functional domain extrinsic to the MIM (residues 151–274, ISP-ED) consisting of 124 residues, containing an $Fe_2S_2$ cluster and located in the mitochondrial inter-membrane space (IMS), an N-terminal trans-membrane (TM) helix that anchors the ISP-ED to the MIM and extends to the matrix part of the Complex III, and a conserved neck region connecting the two domains[26–28].

Incorporation of the ISP into Complex III begins with importing the nuclear-encoded ISP into the matrix as an unfolded precursor polypeptide via the mitochondrial TOM and TIM23 translocases[29], which then refolds into the apo-ISP precursor in the matrix followed by insertion of the $Fe_2S_2$ cluster into the folded ISP-ED (Supplementary Fig. 1D). The translocation of the fully assembled ISP-ED, not the entire ISP precursor, across the MIM and its subsequent incorporation into the core assembly of Complex III are dependent on the function of the AAA protein Bcs1. This ISP biogenesis process is obligatory because the cellular apparatus for iron-sulfur cluster synthesis is found only in the mitochondrial matrix[30].

Bcs1, cyt $bc_1$ synthesis protein 1, was first recognized in yeast as a protein factor required for incorporation of Rip1 (yeast ISP) into cyt $bc_1$ in mitochondria[31,32]. The human ortholog of yeast *bcs1*, *BCS1L* or *Bcs1-like* gene, was identified by a comparative search of genetic factors relating to various mitochondrial diseases[33–36]. Bcs1L is a membrane-bound oligomer, with each subunit consisting of an N-terminal TM segment, a middle Bcs1-specific domain, and a C-terminal AAA domain that is highly distinct, forming a clade of its own in the broad AAA protein family[37]. Genetic and biochemical analyses led to the proposal that Bcs1 enables the assembly of mitochondrial cyt $bc_1$ by translocating the ISP-ED of a fully assembled ISP across the MIM[23].

Structures of yeast Bcs1 and mouse Bcs1L (hereafter referred to as mBcs1L) determined by cryo-EM and X-ray crystallography were found to be very similar. Distinct conformations were reported for Bcs1 associating with different nucleotide-bound forms: apo, ADP, AMP-PNP, and ATPγS[38,39]. All of the reported structures were heptameric, which resolved a long-standing controversy regarding the oligomeric state of Bcs1[40]. The structures of apo and ADP-bound forms appeared to be similar, featuring two cavities, one, located in the matrix, is encircled by the AAA and Bcs1-specific regions, and another, located in the membrane, is bound by seven ordered TM helices. Each cavity is sufficiently large to accommodate a fully folded ISP-ED (Supplementary Fig. 2A). The two cavities are separated from each other by a continuous seven-bladed protein layer made of β-sheet I, forming the Matrix-seal. Characteristic to the Apo/ADP form are the interstitial gap between the Bcs1-specific and AAA regions and a very large opening of 40 Å diameter to the matrix cavity. By contrast, the structures of Bcs1 with bound ATP analogs showed a dramatically contracted conformation with the matrix cavity collapsed to the size one-third of the apo/ADP form and a disordered membrane cavity (Supplementary Fig. 2B). The collapsing of the matrix cavity is accompanied by disappearance of the interstitial gap and reduced opening (20 Å diameter) to the matrix cavity. These structural features are consistent with the proposed function of Bcs1.

Despite the great advancement afforded by these structures, mechanistic steps regarding how Bcs1 couples ATP hydrolysis with substrate translocation have remained unclear. This is because these structures were obtained in the absence of substrate and in uniformly nucleotide or its analog bound states, being either apo, ADP, AMP-PNP or ATPγS. The debate over whether ATP hydrolysis among the seven subunits of Bcs1 is sequential or concerted during substrate translocation remains contentious. Recently, this issue was approached by the

high-speed atomic force microscopy and line scanning (HS-AFM-LS) method in real time and it was found that subunits of mBcs1L hydrolyze ATP in a concerted manner within the detection limit of the method (270 μs)[41].

In this work, we seek to resolve this issue from a different angle by capturing Bcs1 conformations and the nucleotide states for each individual subunit during its active translocation cycle using cryo-EM. We incubate isolated wild-type mBcs1L with ATP in the absence or presence of substrate before performing cryo-EM analysis. Our results show that Bcs1L transitions directly between the ATP and ADP conformations, with no indication of the existence of additional intermediate forms, supporting a concerted mechanism. We also obtain a mBcs1 structure with the ISP-ED trapped in the proposed substrate-binding cavity in the ADP-bound form. Furthermore, the structure of Apo mBcs1L, when reconstructed without imposed symmetry, displays asymmetric arrangement of subunits, suggesting a potential mechanism for substrate ISP precursor release (Supplementary Fig. 1D). The concerted translocation mechanism by Bcs1 contrasts the sequential mechanism employed by most AAA ATPases and sets the stage for complete depiction of the translocation process of ISP.

## Results

### Stimulated ATPase activity and binding of wildtype mBcs1L by substrate ISP-ED

Purified wildtype mBcs1L is active, with a turnover rate near 42 ATP hydrolyzed/Bcs1 heptamer/min in buffer solutions or just under 1 ATP/ heptamer/second (Supplementary Fig. 3A). To test whether the ATPase activity of mBcs1L is modulated by the ISP substrate, we used the extrinsic domain of ISP (ISP-ED), since the ISP-ED of Rip1 was shown to bind Bcs1[23]. The sequences of mature ISP are highly conserved among human, cattle, mouse, and yeast (Supplementary Fig. 1B) and the sequence identity between cattle and mouse ISP-ED is 97%. Using an established method[42], we purified ISP-ED from bovine heart mitochondrial cyt $bc_1$. The purified ISP-ED has an expected molecular weight of 14.4 kDa confirmed by SDS-PAGE and LC-MS (Supplementary Fig. 3B, C), representing a fragment from residues V146 to G274 in the precursor or V68-G196 of the mature ISP (Supplementary Fig. 1B). The light straw color of the isolated ISP-ED solution (Supplementary Fig. 3D) and the crystallization result described in the literature[43] both indicate that the $Fe_2S_2$ cluster is preserved in the purified ISP-ED using this protocol. The ATPase activity of mBcs1L was tested in the presence of purified ISP-ED at different molar ratios between ISP-ED and mBcs1L heptamer (ISP:Bcs1) (Fig. 1A, Supplementary Table 1). Clearly, the ATPase activity of mBcs1L is stimulated by ISP-ED and the maximal stimulation is ~50% at a ISP:Bcs1 ratio of 3-4. Increasing the ISP/Bcs1L ratio further did not seem to affect mBcs1L's activity.

We further tested the binding of isolated ISP-ED with mBcs1L using a pull-down assay. ISP-ED was incubated with Ni-NTA resin with or without hexahistidine-tagged mBcs1L. After washing, proteins retained in the resin were eluted. Sample fractions taken at different steps were analyzed by SDS-PAGE and Western blot using specific anti-ISP and anti-histidine-tag antibodies (Fig. 1B). In the absence of mBcs1L, ISP-ED was washed off completely from the Ni-NTA resin, as shown by the progressively diminishing band intensities in the fractions from the flow through (FT) to the wash and the elution. However, in the presence of mBcs1L, the amount of ISP was enriched on Ni-NTA resin and co-eluted with mBcs1L, indicating an interaction between ISP-ED and mBcs1L. This result is consistent with a previous experiment carried out with yeast Bcs1 and Rip1[23].

### Two conformations captured under ATP hydrolysis conditions in the absence of substrate

The sequential and the concerted mechanisms differ substantially, as the former calls for different nucleotides simultaneously bound

within the heptameric mBcs1L and the latter mandates uniform nucleotide states. Therefore, we decided to use cryo-EM to capture the type of nucleotide bound and the associated conformation of each protomer of mBcs1L while the protein is undergoing active ATP hydrolysis. Similar experiments have been performed with various AAA proteins, and without exception all tested AAA proteins, when substrates were present, were captured with co-existing different nucleotide states (Apo/ADP/ATP) in protomers that arrange in non-planar formations[3–20].

Bcs1 has a basal ATPase activity comparable to that of other AAA ATPases such as p97, which was shown to contain different nucleotides in the native state in the absence of substrate for the D1 domain[44–46]. In the presence of substrate, p97 follows the sequential mechanism with its subunits arranged in a spiral staircase and bound with different nucleotides[4,9]. Because mBcs1L has a relatively slow rate of ATP hydrolysis, which is on average 40-60 ATPs/mBcs1L(heptamer)/min in the absence of substrate, it is conceivable that, if Bcs1 follows the sequential mechanism, stable intermediate states characterized by different conformations and bound nucleotides in different subunits may be able to accumulate. The predominant form of Bcs1 in solution was shown to be ADP-bound after incubation with 1 mM ATP on ice for 60 min[39]. Thus, to capture possible additional major conformational

intermediates, we took the approach of vitrifying sample grids of mBcs1L while it is undergoing active ATP hydrolysis. The sample was prepared by adding near physiological amounts of ATP and $MgCl_2$ to an assay mix containing purified, fully active mBcs1L, and allowing it to proceed for ~10 s. By limiting the time interval between adding ATP/$Mg^{2+}$ and vitrification to only 10 s, various major mBcs1L conformations should be observed, if they exist, with ATP hydrolysis running at an initial rate without too much ADP accumulation. Two possible scenarios are considered here in the absence of substrate (Fig. 2A). If mBcs1L subunits hydrolyze ATP around the ring in a sequential manner, we expect to find minimally varying occupancies of ATP/ADP coexisting among different subunits, as seen in p97[44–46], with additionally subunits arranged in a spiral. By contrast, only uniform, symmetric ADP and/or ATP states should be detected if all seven protomers hydrolyze ATP in a synchronized manner. The experimental conditions adopted here ensured that the ATP hydrolysis reaction was kept active, as the input ATP remained sufficient by the end of the 10 s period.

From 6,912 movies acquired, 997,041 particles were picked automatically after motion corrections and CTF estimations. In reference-free 2D class averages, side and top views of mBcs1L single heptamers were the predominant particles, although tetradecamers, the head-to-head dual heptamers, were also found (Supplementary Fig. 4). We only used the classes corresponding to single heptamers for subsequent processing, as the tetradecamers were known to form through non-specific interactions[38]. Initial 3D reconstruction led to a map at 3.49 Å resolution when no symmetry was imposed (C1), or at 2.93 Å resolution with an imposed C7 symmetry (Supplementary Fig. 4). Further 3D classifications to look for different conformations yielded only two distinct conformations, with 3/4 classes (322,765 particles, approximately 95% of the selected particles after the 2D classification) resembling the ATPγS state with disordered TM cavity (ATP state-1), and a minor population (5%, 15,083 particles) resembling the Apo/ADP state. A 3D reconstruction of the minor class of particles with imposed C7 symmetry yielded a map at 9.55 Å resolution (Supplementary Fig. 4), which can fit a mBcs1L model in the apo/ADP conformation that features an interstitial gap between the AAA and Bcs1-specific region and a large substrate binding cavity or matrix cavity, both of which are hallmarks of the apo/ADP state (Supplementary Fig. 2A). Although the resolution is insufficient to allow direct visualization of bound nucleotide in this minor population, it is nevertheless reasonable to assume that they are in the ADP-bound form based on its overall features. Thus, the detection of a very small fraction of mBcs1L particles in the ADP state and the majority in the ATP state suggests the absence of additional mBcs1L conformations under the experimental conditions.

Another smaller EM dataset was collected with similar conditions but using a different grid type, leading to a uniform ATP-bound reconstruction at 3.46 Å (C1) and 3.02 Å (C7) resolution, respectively. Surprisingly, this map has the membrane cavity better ordered with the TM helices resolved (ATP state-2, Supplementary Fig. 5 and Supplementary Fig. 6A). From this data set, 3D classification did not yield an ADP-state map due to insufficient number of particles, neither were intermediate states of Bcs1L in spiral conformations identified. Although there are differences between ATP state-1 and ATP state-2 (discussed later), the findings from the two independent datasets are consistent, suggesting Bcs1 subunits hydrolyze ATP in a concerted manner.

## ATP occupies all subunits in the ATP-bound forms of mBcs1L
As the map resolution for the apo/ADP form of mBcs1L in the minor class only allowed visualization of overall conformation but not the details in each individual nucleotide-binding pocket, we focused on the 3D reconstruction of particles in the major class for ATP state-1. Using the best class of particles after 3D classification (Supplementary

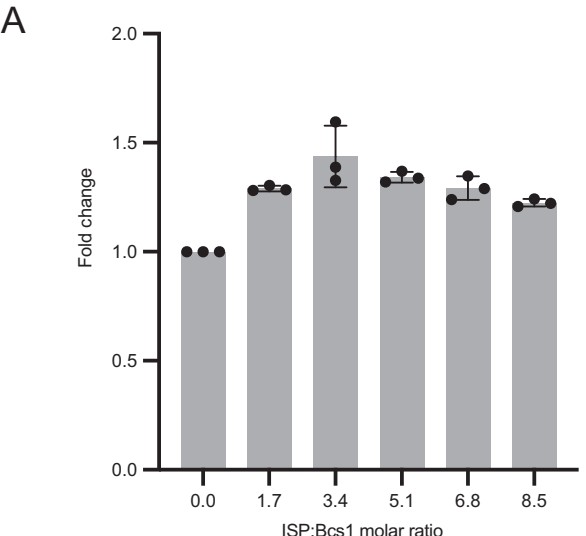

A

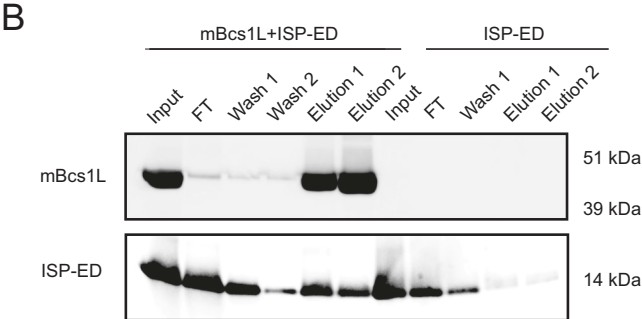

B

**Fig. 1 | Binding of mBcs1L to ISP-ED and elevated ATPase activity. A** mBcs1L ATPase activities in the presence of ISP-ED at different ISP:mBcs1L heptamer ratios. Data shown are mean ± SD of three independent experiments (*n* =3). Source data are provided in Supplementary Table 1. **B** Western blot showing pulldown of ISP-ED by Ni-NTA resin only in the presence of His-tagged mBcs1L. ISP-ED was mixed with Ni-NTA resin in the presence or absence of histidine-tagged mBcs1L. Samples were taken following wash and elution steps and were analyzed by SDS-PAGE and Western analyses. Uncropped scans of blots are provided in the Source Data file.

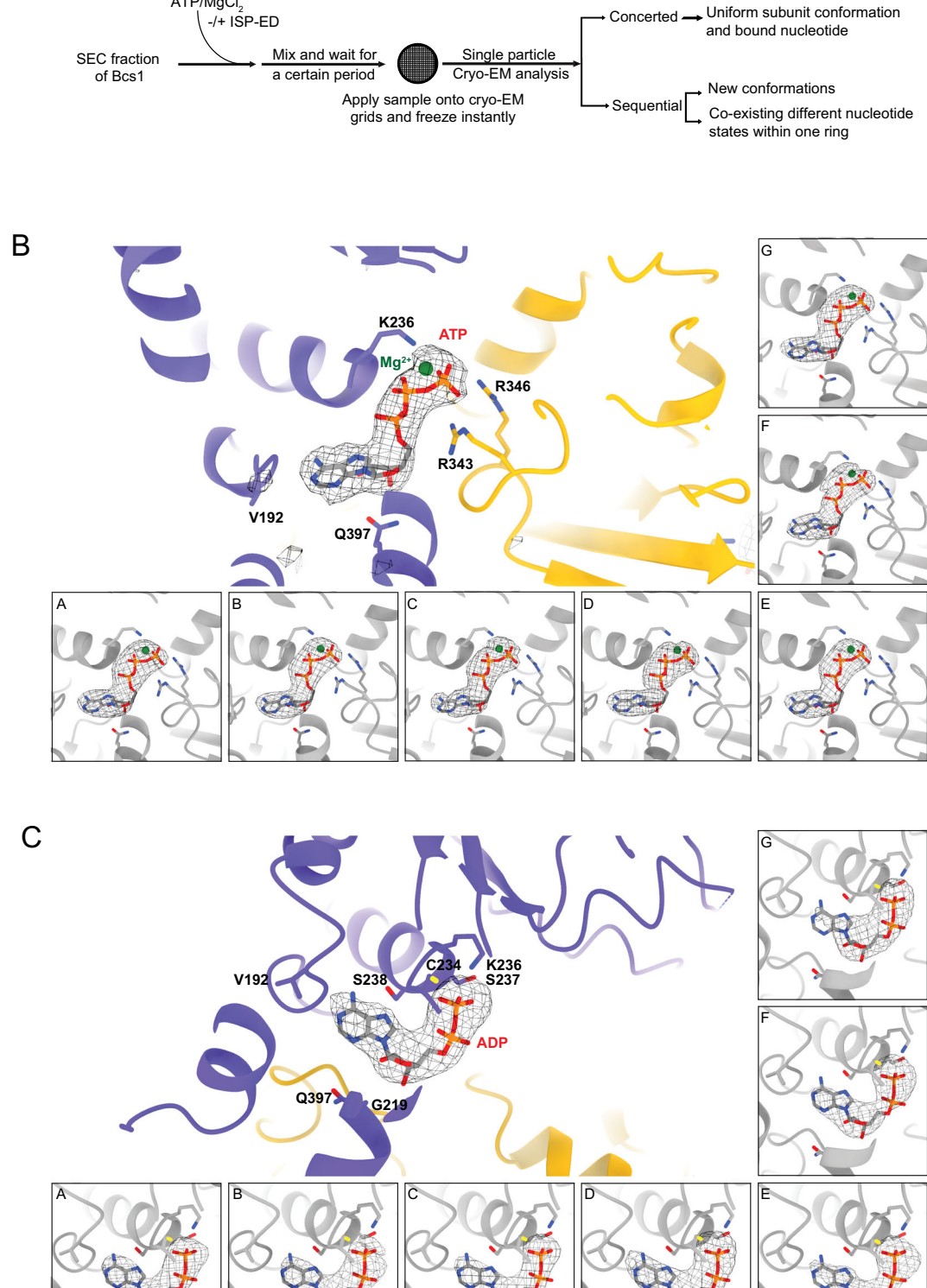

**Fig. 2 | Difference density maps for bound nucleotides in mBcs1L in the ADP and ATP states. A** Experimental procedure used to prepare EM grids. **B** Depiction of bound ATP-Mg²⁺ at nucleotide-binding sites for individual subunits of the structure of ATP-bound mBcs1L determined with C1 symmetry. For each panel, the protein is shown as a ribbon diagram, the bound ATP is rendered as a stick model, and the Mg²⁺ is given as a green ball. Difference EM density generated in Phenix omitting ATP is shown as a gray wire cage and contoured at 15σ level. The middle large panel shows the density of the ATP molecule after the seven-fold symmetry was imposed. **C** Difference EM density shown as wire cages and contoured at 10σ level for the ADP molecules bound at each individual nucleotide-binding site in the structure of mBcs1L determined with C1 symmetry in the presence of ATP and ISP-ED. The middle large panel shows the density of the ADP molecule after the seven-fold symmetry was imposed.

Fig. 4, class 3 in pink), which contained 200,466 particles and did not give detectable tetradecamer signals, the map was refined to 3.77 Å resolution with C1 symmetry and to 3.13 Å with C7 symmetry (Table 1 and Supplementary Fig. 4). The overall shapes of the two reconstructed maps with and without imposed symmetry looked the same, both closely resembling the mBcs1L-ATPγS conformation (PDB:6UKS, Supplementary Fig. 2A). Global superposition of the refined mBcs1L-ATP heptamer model with the reported model of mBcs1L with bound ATPγS gave a rms deviation of 0.644 Å for 1,743 aligned CA atoms (Supplementary Table 2) with the ATP-bound Bcs1L slightly expanded in diameter. Close-up examination of the EM density map constructed with C1 symmetry confirmed without exception that all seven nucleotide-binding pockets are occupied by ATP • Mg$^{2+}$ (Fig. 2B). In both C1 and C7 maps, the nucleotide-binding sites are exceptionally well-defined in the density, with local resolutions estimated at 3.5 and 3.0 Å, respectively, exceeding the global resolutions of the reconstructed maps and providing confidence in the assignment of nucleotide state (Supplementary Fig. 4C). Consistently, all seven subunits of mBcs1L display a uniform conformation, remaining as a planar ring. Identical uniform ATP binding was also observed for mBcs1 in ATP state-2.

Compared to the ATPγS bound mBcs1L (PDB:6UKS), the AAA ATPase region of mBcs1L in ATP state-1 superposed very well, whereas its Bcs1-specific region aligned poorly (Supplementary Fig. 6B). By contrast, mBcs1 in ATP state-2 aligned both its AAA ATPase and Bcs1 regions well (Supplementary Fig. 6C). This difference is reflected in the global superpositions (Supplementary Table 2), where only 1,743 residues were aligned for the ATP state-1, whereas for ATP state-2, the number of aligned residues is 2,338. Also noticeable in ATP state-2 is the ordered TM region compared to mBcs1L in ATP state-1 and in the ATPγS bound state.

## Nucleotide states and conformations of mBcs1L captured in the presence of ATP and substrate ISP-ED

Since the spiral arrangement of subunits of canonical AAA proteins has always been observed in the presence of substrates[3–16], one could argue that the observed uniform ADP or ATP conformations of mBcs1L were a consequence of the ATP hydrolysis reaction in the absence of substrate, which could change when mBcs1L actively transports ISP across the MIM.

Our pull-down experiments showed a direct interaction between mBcs1L and bovine ISP-ED (Fig. 1B), which justified the use of bovine ISP-ED as a model substrate for capturing mBcs1L conformation while it is undergoing ATP hydrolysis cycles. Purified mBcs1L was mixed with ISP-ED at a molar ratio of 1:1 (mBcs1L heptamer:ISP) and incubated on ice for about 30 min in the presence of 2 mM ATP and 20 mM MgCl$_2$ before vitrification. 3D classification and refinement again yielded only two main conformations corresponding to the ATP- and ADP-bound state, respectively. No spiral conformations were identified in either the 2D or 3D classes. Among the three selected final classes (Supplementary Fig. 7), Class 1 (in magenta with 13,851 particles or 23%) resembled ATPγS conformation, while Class 3 (in green with 8,300 particles or 14%) and Class 4 (in blue with 37,681 particles or 63%) resembled Apo/ADP conformation. The Class 3 reconstruction deviated from the previously established ADP state by the presence of an extra piece of globular density in the matrix cavity, which was not observed in the Class 4 reconstruction. Using all particles in Class 4, the ADP state map reached a resolution of 3.74 Å when no symmetry was applied (C1), and the resolution improved to 3.18 Å when the C7 symmetry was used (Table 1, Supplementary Fig. 7)

Inspection of the nucleotide-binding sites in the unaveraged map for Class 4 particles found densities that could only be fit with ADP for all seven subunits (Fig. 2C). The quality of the density for the bound ADP was further enhanced in the map with applied C7 symmetry in the final refinement and reconstruction. Again, no mixed nucleotide-

binding states were observed, suggesting that subunits of mBcs1L also hydrolyze ATP in a concerted manner in the presence of ISP-ED. The ADP-bound mBcs1L structure has an appearance similar to that of the previously reported apo mBcs1L structure (PDB:6UKP)[38]. Indeed, structure alignment between the two heptamers gave rise to a rms deviation of 1.058 Å for 2,278 superposed CA atoms (Supplementary Table 2).

## Extra density the size of ISP-ED found only in the Class 3 map

Although the 3D classification and refinement only yielded two distinct conformations corresponding to an ATP state (23%) and ADP state (77%), respectively, the ADP state could be further divided into 2 classes, one with extra density in the matrix cavity (14%) and one without (63%). This extra piece of density lodged in the putative substrate-binding cavity is not likely to be a part of mBcs1L, as it does not align with the 7-fold axis of mBcs1L ring and is asymmetrically distributed with contacts to some but not all the mBcs1L subunits (Fig. 3A). At the 7.2 Å resolution, this density fits the size of an ISP-ED model (PDB:1RIE, Fig. 3B) with a log-likelihood gain of 116 and a model-map correlation coefficient of 0.64. The reconstruction fails to reach higher resolution, probably due to insufficient number of particles. The low percentage of particles with ISP-ED occupied (14% of the total particles used in the reconstruction) could be the result of ISP-ED being constantly translocated by mBcs1L, while actively hydrolyzing ATP or due to relatively low binding affinity.

It should be noted that this additional density persists when the same data set was processed using different EM data processing software such as cisTEM and cryoSPARC in parallel, ruling out the possibility of a software-dependent artifact. Processing of Class 4 particles in an identical fashion did not yield the extra density in the substrate-binding cavity, nor did the processing of Class 1 particles in the ATP conformation (Fig. 3A). Furthermore, we re-processed the apo mBcs1L data set (EMD-20808) with imposed C1 symmetry to 4.4 Å resolution (Fig. 3A). No such globular density was observed in either lowpass filtered or unfiltered maps.

Taken together, our results strongly indicated that ISP-ED is capable of interacting with the mBcs1L heptamer in the ADP state. This interaction appears to be mediated via one or two of the seven equivalent subunits. Since the Bcs1 Apo state closely resembles the ADP state in conformation and has been shown to interact with the ISP-ED (Fig. 1B), it is reasonable to assume that the ISP-ED could only enter the substrate binding pocket in the Apo/ADP state, which likely represents a pre-translocation state in which substrate is bound but ATP is yet to enter the nucleotide binding pockets to induce a conformational change. The mBcs1L-ADP structure determined here was captured under active ATP-hydrolyzing conditions and in the presence of the ISP-ED substrate. The fact that no co-existing nucleotide states are found while mBcs1L is undergoing active ATP hydrolysis in the presence or absence of substrate strongly advocates for a concerted mechanism.

## Apo Bcs1L displays asymmetrical subunit arrangement

Previously, we reported the structure of mBcs1L in the absence of nucleotide and substrate (Apo mBcs1L) determined by cryo-EM with imposed C7 symmetry[38]. When we imposed no symmetry (C1) in reprocessing this data set (Supplementary Fig. 8), the resulting EM map at 4.4 Å resolution showed significant asymmetrical arrangement of mBcs1L subunits in the AAA ATPase domain with appearance of large crevices between some subunits (Fig. 4A). Inspections of the model built into the map confirmed that the separations between neighboring subunits in the AAA domains are not uniform. In particular, the separation between subunits A and G is clearly wider than others (Fig. 4B). This asymmetry is also reflected in the rms deviation of 1.222 Å for 2,259 residues when the C1 model is superposed to the C7 model (Supplementary Table 2), which is rather large compared to

**Table 1 | Cryo-EM data collection, model refinement and validation statistics**

| | ATP state-1 | | ATP state-2 | | ADP | | | Apo |
|---|---|---|---|---|---|---|---|---|
| ISP-ED present in solution | No | | No | | Yes | | | No |
| **Data collection** | | | | | | | | |
| Microscope and camera | Titan Krios, K2 | | Titan Krios, K3 | | Titan Krios, K3 | | | Titan Krios, K2 |
| Magnification | x29,000 | | x105,000 | | x105,000 | | | x14,000 |
| Voltage (kV) | 300 | | 300 | | 300 | | | 300 |
| Frames/micrograph | 40 | | 50 | | 50 | | | 38 |
| Electron exposure ($e^-/Å^2$) | 50 | | 54.4 | | 54.4 | | | 40 |
| Defocus range (μm) | 1–2.5 | | 0.7–2.0 | | 0.7–2.0 | | | 1.25–2.5 |
| Pixel size (Å/pixel) | 0.858 | | 0.415 (0.83 binned) | | 0.415 (0.83 binned) | | | 0.858 (1.72 binned) |
| Total no. of micrographs | 6912 | | 1839 | | 6900 | | | 2314 |
| **Data processing** | | | | | | | | |
| Reconstruction software | cisTEM | | CryoSparc | | CryoSparc | | | CryoSparc |
| Extracted particles (no.) | 997,041 | | 33,415 | | 973,807 | | | 1,430,475 |
| Selected 2D particles (no.) | 337,848 | | 28,638 | | 72,693 | | | 212,961 |
| Particles for final map (no.) | 200,466 | | 26,234 | | 37,681 | | 8,300 | 212,961 |
| Starting model | De novo | | De novo | | De novo | | De novo | De novo |
| ISP-ED found in map | - | - | - | - | No | No | yes | - |
| Symmetry imposed | C1 | C7 | C1 | C7 | C1 | C7 | C1 | C1 |
| Map resolution (Å)[a] | 3.77 | 3.13 | 3.46 | 3.02 | 3.74 | 3.18 | 7.2 | 4.40 |
| Map resolution range (Å) | 3–8 | 2.5–4.5 | 3–7 | 2.5–6.5 | 9.5–3.5 | 6.0–3.0 | | 9–4.0 |
| Map sharpening B-factor ($Å^2$) | –90 | –90 | –59 | –96 | –62 | –97 | –293 | –180 |
| **PHENIX refinement statistics** | | | | | | | | |
| Initial model used | 6UKS | | 6UKS | | 6UKO | | | 6UKO |
| Model composition | | | | | | | | |
| Non-hydrogen Atoms | 20,580 | 20,580 | 23,702 | 23,702 | 22,799 | 22,722 | 23,696 | 21,126 |
| Residues | 2520 | 2520 | 2933 | 2933 | 2821 | 2814 | 2937 | 2618 |
| Ligands | 7Mg$^{2+}$, 7ATP | 7Mg$^{2+}$, 7ATP | 7Mg$^{2+}$, 7ATP | 7Mg$^{2+}$, 7ATP | 7ADP | 7ADP | 7ADP 1FES | 0 |
| Overall B-factor ($Å^2$) | | | | | | | | |
| Protein | 211.5 | 178.3 | 135.2 | 145.1 | 163.3 | 155.1 | 377.5 | 207.0 |
| Ligands | 170.8 | 136.6 | 87.5 | 100.5 | 169.8 | 158.5 | 347.1 | |
| RMSD deviations | | | | | | | | |
| Bond length (Å) | 0.003 | 0.002 | 0.002 | 0.004 | 0.002 | 0.002 | 0.003 | 0.004 |
| Bond angle (°) | 0.608 | 0.539 | 0.438 | 0.545 | 0.586 | 0.572 | 0.741 | 0.928 |
| FSC model vs. map = 0.5. (Å) | 4.48 | 3.46 | 3.84 | 3.26 | 4.27 | 3.55 | | 4.80 |
| Real-space correlation | | | | | | | | |
| CC (mask) | 0.82 | 0.86 | 0.87 | 0.89 | 0.83 | 0.88 | 0.65 | 0.83 |
| CC (volume) | 0.81 | 0.85 | 0.85 | 0.88 | 0.82 | 0.87 | 0.58 | 0.82 |
| CC (peaks) | 0.67 | 0.69 | 0.73 | 0.74 | 0.65 | 0.68 | 0.39 | 0.64 |
| Mean CC for ligands | 0.93 | 0.92 | 0.92 | 0.91 | 0.84 | 0.84 | 0.71 | - |
| **Validation** | | | | | | | | |
| MolProbity score | 1.67 | 1.64 | 2.02 | 1.87 | 1.32 | 1.63 | 2.13 | 1.73 |
| Clash score | 6.78 | 6.17 | 7.80 | 10.44 | 5.84 | 7.81 | 16.53 | 11.05 |
| Rotamer outliers (%) | 0.18 | 0.23 | 2.52 | 0.00 | 0.00 | 0.00 | 0.00 | 0.00 |
| Ramachandran plot | | | | | | | | |
| Favored (%) | 95.7 | 95.7 | 95.8 | 95.2 | 98.0 | 96.7 | 93.9 | 97.0 |
| Allowed (%) | 4.0 | 4.0 | 4.2 | 4.8 | 2.0 | 3.3 | 5.9 | 3.0 |
| Disallowed (%) | 0.3 | 0.3 | 0.0 | 0.0 | 0.0 | 0.0 | 0.2 | 0.0 |
| **Data deposition** | | | | | | | | |
| PDB | 8TI0 | 8T5U | 8TPL | 8TP1 | 8T7U | 8T14 | | 8TBY |
| EMD | 41276 | 41061 | 41476 | 41462 | 41095 | 40954 | 41609 | 41148 |

[a]map resolution was based on FSC threshold of 0.143.

those identically calculated with the ATP- and ADP-bound structures of 0.285 Å and 0.248 Å, respectively. The apparent weak AAA domain association of mBcs1L subunits in the apo state is supported by the calculated buried surface area (BSA) between pairs of neighboring subunits, with lower BSA indicating weaker associations. In particular, the BSA between subunits A and G is the smallest at 4317 $Å^2$ (Supplementary Table 3). Calculations of BSA for ADP- and ATP-bound mBcs1L showed on average ~1000 $Å^2$ more buried surface area than that for apo mBcs1L, suggesting that nucleotide binding stabilizes oligomeric states. Similar observations were reported for AAA proteins ClpA and p97, in which subunits in the apo state are most disordered[47,48].

To further validate this observation, we expressed the AAA domain of mBcs1L ($^{ΔN}$mBcs1L, residues 151–420) and tested its oligomeric state in the presence of various nucleotides using BN-PAGE (Fig. 4C). Under the conditions of no nucleotide (Apo) or in the presence of ADP, $^{ΔN}$mBcs1L does not form heptamers, but rather exists as

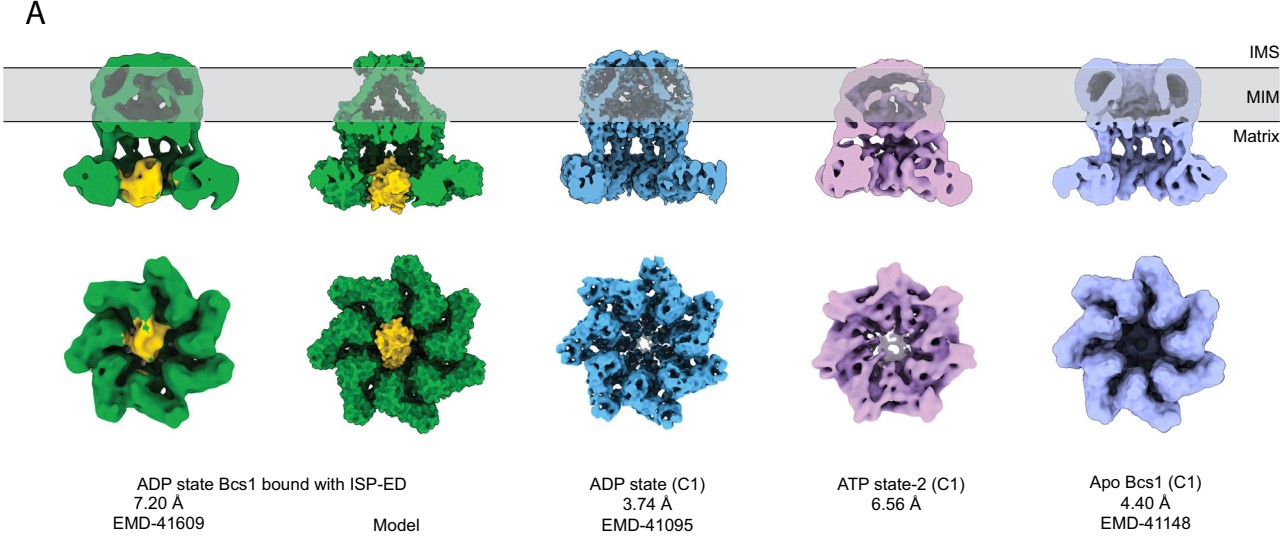

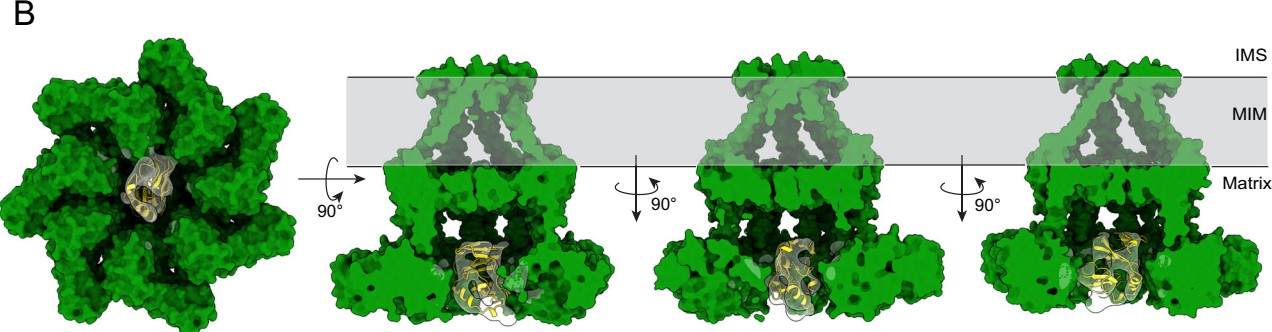

**Fig. 3 | Substrate ISP-ED binding to the substrate-binding cavity is detected only in the ADP state. A** Three final classes from the mBcs1L/ISP-ED data set are shown in cut-away view (top) exposing the interior content of the substrate-binding cavity and the view from the bottom (bottom). From left to right: experimental map of mBcs1L (green) bound with ISP-ED (yellow); surface rendition of the mBcs1L model (green) and modeled ISP-ED (yellow); the experimental map of ADP-bound mBcs1L without ISP-ED (Blue); the experimental map of mBcs1L in ATP state-2 (magenta); and the map from apo mBcs1L (purple) as a control. All reconstructions were performed in C1 symmetry. **B** Surface representation of mBcs1L bound with ISP-ED shown in different orientations. A model of ADP-state mBcs1L was fit into the map using ChimeraX, followed by fitting a model of bovine ISP-ED (1rie.pdb) into the extra density in the matrix cavity using the Emplace_local tool via ChimeraX. The sharpened local map generated by Emplace_local is shown as semi-transparent surface.

monomers, dimers, or trimers. By contrast, in the presence of non-hydrolyzable ATP analogs such as AMP-PNP and ATPγS, heptameric association of ᴬᴺmBcs1L subunits is strongly favored.

When measuring the molecular weight of recombinantly expressed mBcs1L using mass spectrometry, we also noticed that apo mBcs1L dissociated and could be detected as monomers with a molecular weight of 48 kDa in the absence of nucleotide (Supplementary Fig. 9A). By contrast, in the presence of ATPγS, no monomer signal was detected, consistent with the formation of tight heptamers that would be beyond the detectable molecular mass range of the instrument (Supplementary Fig. 9B). This data agrees with the notion that subunits of mBcs1L may dissociate when it is in the apo state.

**Structure of mBcs1L in the TM region**
To obtain the best resolution for the TM region of mBcs1L structures, we performed 3D reconstructions in C7 symmetry for the two ATP-bound maps (ATP state-1 and ATP state-2) and the ADP-bound map (Supplementary Figs. 4, 5 and 7). These averaged maps reached higher resolution and allowed more structural features to be visualized in both ADP- and ATP-bound states. For example, in our ADP and ATP state-2 maps, we could trace the density all the way to the N-terminus of mBcs1L, depicting the structures of the entire N-termini exposed to the IMS to form the IMS seal, the complete TM helices, and additionally

the apparent ordered densities that could be modeled as lipid molecules circling the base of the membrane basket (Fig. 5 and Supplementary Fig. 10). These structural features were not visible in the previous reports[38,39].

In the ADP state, the first 11 residues at the N-terminus (residues M1 to K11) form a 3-turn short helix, which lies parallel to the surface of MIM and is connected by a short loop to the 50 Å long, 9-turn TM helix (residues P14 to Y47) that spans the MIM (Fig. 5A, left). This N-terminal assembly consisting of the 3-turn short helix, the connecting loop, and the beginning (¹⁴PYF¹⁶) of the long TM helix makes contacts with their counterparts of neighboring subunits to form a seal on the IMS side of the membrane, hereafter referring to as IMS-Seal (Fig. 5B, left). The IMS-Seal in the ADP state features a narrow opening of ~9 Å in diameter, which is constructed by the side chains of Y15 pointing up and towards the center.

From the IMS-Seal, the TM helices radiate down toward the matrix side of the membrane, separating from each other by 30 Å on the matrix side of the MIM and giving the TM region of the mBcs1 the shape of an up-side-down basket, as previously described for the yeast Bcs1 (PDB:6SH3)[39]. Herein, we visualize the IMS-Seal structure. The yeast Bcs1 structure in the ADP-state (PDB:6SH3) starts at residue N49, missing 48 N-terminal residues. Since the yeast Bcs1 has a longer N-terminus, its first residue, N49ˢᶜ, corresponds to N13ᴹᴹ of mBcs1L.

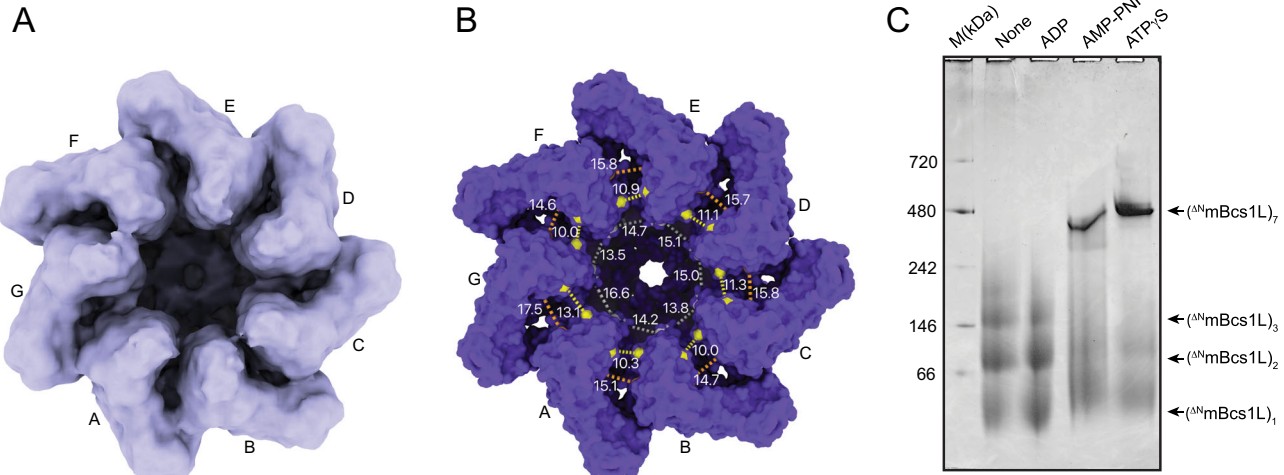

**Fig. 4 | Apo state of mBcs1L displays asymmetric subunit arrangement. A** View from the matrix side of the mBcs1L experimental map in apo state reconstructed in C1 symmetry. **B** Surface representation of the apo mBcs1L model built into the map. The separations between neighboring subunits in the AAA region are indicated by the distances measured between three pairs of atoms: gray dashed line is between $^{CD}R177^X$ and $^{CB}E175^{X+1}$, yellow dashed line is between $^{CA}D258^X$ and $^{CB}D264^{X+1}$, and the orange dashed line is between $^{CA}D283^X$ and $^{CB}N314^{X+1}$, where X is chain id. **C** BN-PAGE analysis of $^{ΔN}$mBcs1L oligomeric state in the absence (None) or presence of ADP, AMP-PNP, and ATPγS. Representative of $n = 2$ independent experiments. Uncropped scan of the BN-PAGE is provided in the Source Data file.

Interestingly, our C7-symmetrized map reveals considerable amount of unassigned density that encircles the IMS-Seal inside the basket and fills the gaps between TM helices. For example, a phosphatidyl ethanol amine (PE) lipid is modeled into the difference density at the matrix side of MIM (Supplementary Fig. 10).

**Conformational difference between ATP state-1 and ATP state-2**
As mentioned earlier, two independent ATP states were identified in this work: ATP state-1 and ATP state-2. Their overall conformations resemble the previously reported ATPγS-state[38]. The AAA ATPase regions are very much the same, but they differ substantially in the TM region and the Bcs1-specific region when compared to the ATPγS-state and to each other (Supplementary Table 2 and Supplementary Fig. 6). In the ATP state-1, the entire IMS-seal and the TM helices are disordered (Fig. 5A, middle), which has also been reported for the ATPγS-state[38]. By contrast, the ATP state-2 reveals a near-complete structure of the TM region, showing a similar membrane-embedded basket encircled by TM helices as in the ADP state (Fig. 5A, right).

To describe the motion of TM helices as mBcs1L transitioning from the ATP- to ADP-bound states, we define two angles: Ψ and φ. Ψ is a dihedral angle defined by 4 CA atoms: $G35^{chainX}$, $Y47^{chainX}$, $Y47^{chainX-1}$ and $Y47^{chainX+1}$ and φ is a tilting angle between the axis of TM helix$^{chainX}$ and the vector on the membrane surface between two equivalent CA atoms: $Y47^{chainX}$ and $Y47^{chainX+1}$ (X is a chain id, Fig. 5D). Here, Ψ represents the dihedral angle between the plane where the TM helix resides and that of the membrane surface, whereas change in φ represents an in-plane rotation of the TM helices as mBcs1L undergoes the ATP to ADP transition. We observed, by measuring Ψ and φ for mBcs1L in the ATP state-2 and ADP state, that while the dihedral angle Ψ remains constant at 50° during the transition, the tilting angle φ changes substantially. For the ATP state-2, the TM helices pass through the MIM at an apparent tilting angle φ of 60°, leaning down more towards the membrane plane compared to the ADP state that has a tilting angle of 70° (Fig. 5C). This difference in φ results in a shallower TM basket and a slightly more opened IMS-seal with a diameter of ~13 Å (Figs. 5B and 5C), which is consistent with a previous observation by AFM showing the IMS seal undergoing an up-and-down movement during turnover[41]. The EM density for TM region of mBcs1L structure in the ATP state-2 is the least well-defined with reduced local resolution, indicating elevated mobility in this region (Supplementary Fig. 5C). Thus, the transition from ATP state-

2 to ADP and vice versa in the TM helices resembles the opening and closing of an aperture's iris diaphragm to release the ISP-ED into the IMS (Fig. 5B and supplementary movie 1).

Substantial conformational changes have also been observed in the Bcs1-specific region (Fig. 5A). In the apo/ADP states, the membrane basket in MIM is separated from the matrix cavity by the Matrix-seal that is parallel to the membrane plane (Supplementary Fig. 2A). This seal is formed by the β-sheet I, protruding towards the center, with the central seal loops tightly packed together (Fig. 5A, bottom). In the ATP state-2, the β-sheet I tilts away from the MIM, unpacking the seal loops and opening the Matrix-seal. The Matrix-seal in the ATP state-1 has the widest opening, as the β-sheets I tilts further away from MIM and the seal loops become too flexible to trace in the density map (Fig. 5A, middle). Following the changes in the Matrix-seal during ATP/ADP transition, the β-sheet II that connects the Matrix-seal to the AAA domain becomes bent in the ATP state-2 from its fully extended conformation in the ADP state (Fig. 5A), effectively causing the interstitial gap between the Bcs1-specific and the AAA regions to disappear. The conformation of β-Sheet II in ATP state-1 is partially extended with the density at the tip being less well-defined (Fig. 5A, top). The functional relevance of these two observed ATP states remains to be seen.

## Discussion
In this work, we captured conformations and nucleotide states of mBcs1L and its protomers in their translocation competent states and during their active ATPase cycle in the presence or absence of substrate ISP-ED, all of which indicate that mBcs1L translocates ISP precursor in a concerted manner. The concerted mechanism employed by Bcs1 is in sharp contrast to canonical AAA proteins that were found to follow the threading mechanism under similar experimental conditions[3–5]. Our experiments additionally provide structural and biochemical evidence for the interaction of mBcs1L with its physiological substrate ISP-ED (Fig. 3). The trapping of ISP-ED in the matrix cavity by mBcs1L in its ADP-state (Fig. 3) should be considered equivalent to ISP-ED binding to apo mBcs1L, as demonstrated by our pulldown experiment (Fig. 1B), which is consistent with published work with yeast Bcs1 and represents a pre-translocation state[23].

Using similar experimental designs as shown in this work, previous cryo-EM studies of canonical, hexameric AAA ATPases revealed a shared threading mechanism, that is characterized by a dominant

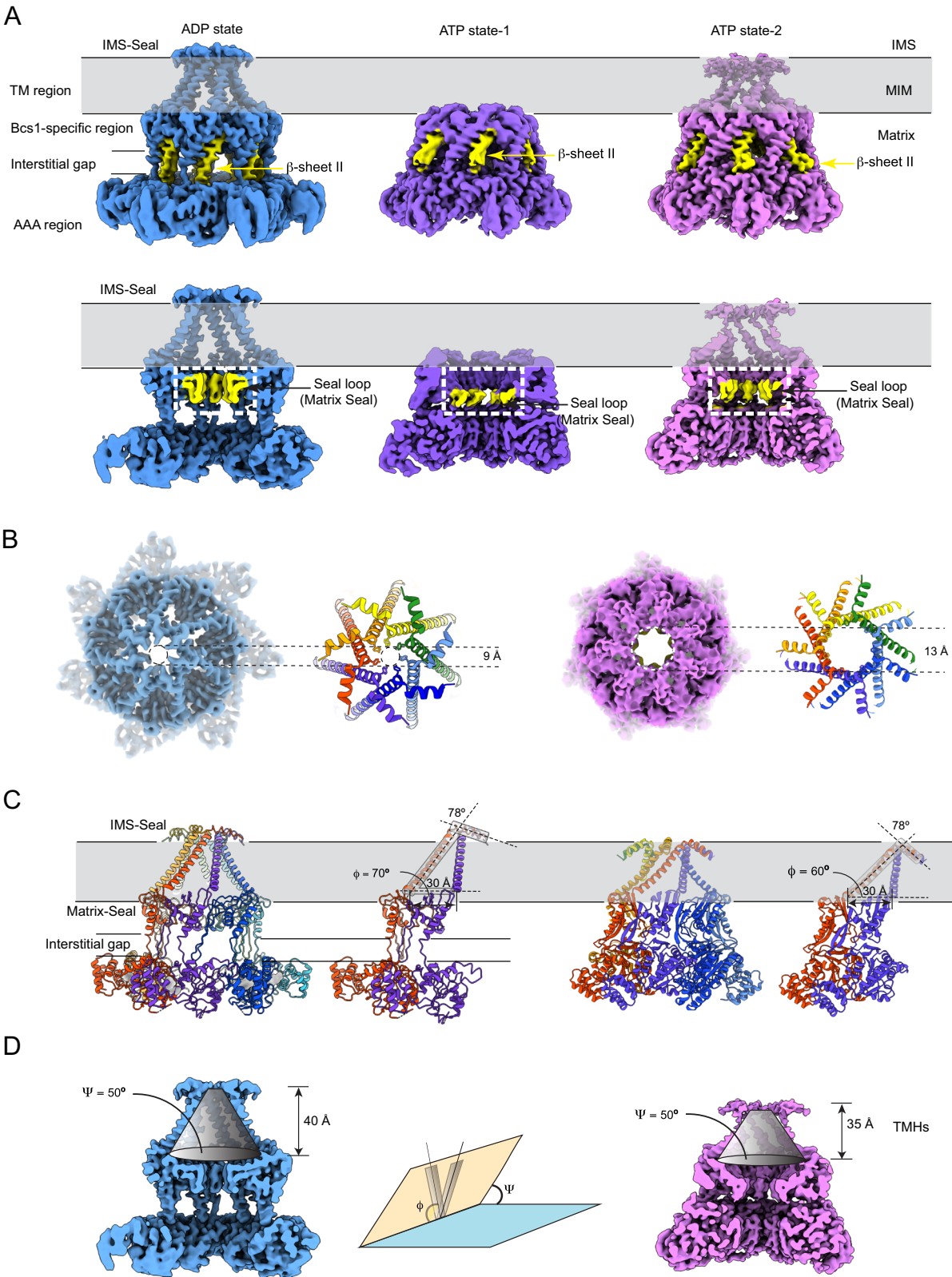

**Fig. 5 | Nucleotide-dependent conformational change in the membrane cavity.**
**A** EM density of the mBcs1L in ADP state (left), ATP state-1 (middle) and ATP state-2 (right). Top panel shows the EM densities in sideview, with the β-sheet II of Bcs1-specific region highlighted in yellow and indicated with yellow arrows. Bottom panel is the cut-away view with the seal loops highlighted in yellow and indicated with black arrows. **B** Top view from the IMS side of mBcs1L map and the model for IMS-seal in ADP state (left) and ATP state-2 (right). The sidechain of Y15 is shown in stick models. **C** Cartoon model of the full-length mBcs1L in ADP state (left) and ATP state-2 (right). A pair of adjacent mBcs1L subunits can be used to define the in-plane angle ϕ of the TM helix. **D** The membrane-embedded basket encircled by TM helices in the ADP state (left) and in ATP state-2 (right). Definition of the in-plane rotation angle (ϕ) and the dihedral angle (Ψ) is shown in the middle.

spiral conformation[3-15]. Five of the six subunits form a descending right-handed spiral around a polypeptide chain or DNA substrate, while the sixth "seam" subunit lies in the middle to bridge the top and bottom of the spiral[49]. Except for the "seam" subunit, the five other subunits in the spiral all have their conserved pore-loops protruding into the central pore in direct contact with the substrate. The nucleotide occupancy for each individual subunit is highly correlated with its position in the spiral, with the top three subunits always bound with ATP and the "seam" subunit in the post-hydrolysis state (Apo/ADP). The two subunits at the bottom of the spiral were frequently seen in the post-hydrolysis state. Substrate translocation is thought to go through sequential ATP hydrolysis, as individual subunit transitions from ATP-bound form to extended ADP-bound form, thus moving downwards while pulling the substrate along stepwise. However, this sequential mechanism faces difficulties with the observations from some AAA proteins like RavA and ClpX, which feature two separate seam subunits in cryo-EM structures, leading to the proposed probabilistic mechanism for ATP hydrolysis[16-20].

These elegant mechanisms for AAA protein functions are incompatible with Bcs1 function. Translocation of folded proteins across lipid bilayers is conceivably more difficult than translocating while unfolding proteins via the threading mechanism. However, both prokaryotic and eukaryotic cells have evolved ways to carry out this task. In photosynthetic bacteria, the ISP is incorporated into the bacterial cyt $bc_1$ complex by the Tat (twin arginine translocation) system[50], which is a general pathway in proteobacteria for folded protein transport across the membrane into the periplasm[51]. This pathway is replaced by Bcs1 in eukaryote mitochondria to specifically translocate ISP-ED into the mitochondrial intermembrane space.

Although the translocation mechanism of ISP-ED by either the Tat system or Bcs1 remains speculative, requirements for this process have emerged. First, the machinery must be sufficiently large to accommodate the folded ISP. Second, the movement of ISP-ED across the membrane is energetically demanding. Third, during ISP-ED translocation, the MIM must remain sealed, avoiding breakdown of the proton motive force across the membrane. Finally, the TM helix and the fused subunit 9 that are still associated with Bcs1 must be released post ISP-ED translocation (Supplementary Fig. 1). Recent and current structural studies have offered important insights into how Bcs1 meets these requirements and are in support of the concerted translocation mechanism[38,39,41].

Unlike canonical AAA proteins with narrow translocation pores, Bcs1 is able to accommodate a large, folded ISP substrate. Bcs1 deviates from canonical AAA proteins in several major aspects, and thus was classified into its own clade[37]. It is heptameric, as compared to the typical hexameric AAA proteins. Additionally, in its highly conserved AAA-ATPase domain, the RecA-like subdomain of Bcs1 has a seven-stranded β-sheet, in contrast to most AAA proteins that feature a five-stranded β-sheet, effectively increasing the size of its substrate-binding cavity. Bcs1 also lacks the conserved pore-loop that is used by canonical AAA proteins to interact with substrates in translocation pore[52].

The energetic cost of transporting ISP-ED across a charged MIM has not yet been measured, but the cost to move a small hydrophobic molecule such as rhodamine 123 by an ABC transporter under optimal conditions was shown to be one ATP (7.2 kcal/mol)[53]. Considering the energy barrier that must be overcome to move the 14.4 kDa ISP-ED across the membrane, the concerted mechanism is more plausible because it can deliver ~50 kcal/mol of energy instantly by simultaneously hydrolyzing all seven ATPs.

How Bcs1 maintains the integrity of the membrane while pushing the folded ISP-ED across the electrically charged mitochondrial inner membrane is not understood. One hypothesis is an airlock-like mechanism afforded by the two compartments[39], with the substrate-binding cavity in the matrix and a basket in the membrane. These two compartments are separated by the "Matrix-seal" in the Bcs1-specific domain (Fig. 5A and Supplementary Fig. 2). An important feature revealed in this work is a second seal, referred to as the "IMS-seal", that is composed of the N-terminal short helices of the Bcs1 subunits and caps the membrane basket (Fig. 5). Although the two seals could provide a mechanism for an airlock like system to operate in Bcs1, our current data do not show alternating opening and closing of the two seals. The need to maintain the membrane integrity is underscored by the observation that all eight mutations (R45C, M48V, T50A, R73C, S78G, P99L, R109W, and R144Q) identified from patients with GRACILE syndrome[38,54], a severe form of Bcs1L-linked mitochondrial disease, are clustered at the border between the TM region and Bcs1-specific region and at the subunit interface (Supplementary Fig. 11).

Finally, the translocated ISP precursor must be released from Bcs1. It was shown biochemically that the N-terminus of Rip1 could interact with yeast Bcs1 in the presence of AMP-PNP, and ATP hydrolysis is required to release Rip1 from Bcs1 in vitro[23]. The structural basis of this process remains speculative, as here the cleaved ISP-ED that lacks the N-terminal TM helix and fused subunit 9 was used as substrate. Nevertheless, structural analysis of mBcs1L in different nucleotide states indicates that the ISP precursor most likely is released in apo state. Bcs1 subunits in apo state show significant asymmetric arrangement with large gaps between some subunits and are more prone to dissociate (Fig. 4, Supplementary Table 3 and Supplementary Fig. 9), which implies a mechanism for ISP precursor release, where Bcs1 subunits open up laterally in apo state to let ISP precursor diffuse out. This is consistent with canonical AAA proteins such as p97 that release the D2 domain of the "seam subunit" after ATP hydrolysis[9]. Similar "side-window" mechanism to allow substrate diffusion in and out of the proteasome 20S core particle has also been proposed[55-57].

Incorporating the herein identified conformational states into the existing Bcs1 mechanistic framework[38], we propose that mBcs1L accomplishes the translocation of ISP precursor in at least five steps (Fig. 6): (1) ISP precursor binds to the apo Bcs1L in the substrate-binding cavity. (2) ATP binding to all seven subunits initiates a concerted conformational change in Bcs1L, opening both Matrix-seal and IMS-seal and pushing the ISP-ED across the membrane, as represented by the ATP state-1 or ATP state-2. (3) Conceivably, in ATP-bound state, the post-translocation state, Bcs1 captures ISP precursor by its TM helix. (4) ATP hydrolysis restores the apo/ADP conformation. (5) During ADP/ATP exchange, one subunit of Bcs1 may partially dissociate from the complex, allowing complete release of the substrate.

## Methods

### mBcs1L expression and membrane isolation

The expression and purification of mBcs1L has been described previously[38], to which we made slight modifications. The cell pellet was resuspended in homogenization buffer (100 mM Tris, pH 8, 100 mM sucrose, 1 mM EDTA, and 2 mM PMSF) before applying to an microfluidizer (Microfluidics International Cooperation) at 1,000 bars for three cycles. Insoluble debris was separated from the supernatant by centrifugation at 3,500 x g for 25 min. Crude membranes in the supernatant were collected by ultracentrifugation at 125,000 x g for 60 min. Subsequently, the membrane pellet was resuspended in the wash buffer (25 mM Tris, pH 8, and 200 mM NaCl) followed by ultracentrifugation at 125,000 x g for 90 min. The final crude membrane pellet was resuspended in the storage buffer (25 mM potassium phosphate, pH 7.4 and 200 mM NaCl) and the total protein concentration was determined using the Pierce BCA Protein Assay Kit (Thermo Fisher Scientific, Waltham, MA).

### Purification of mBcs1L

The total protein concentration in the isolated membrane was adjusted to 5 mg/ml using the storage buffer, and 20% CHAPS (Anatrace)

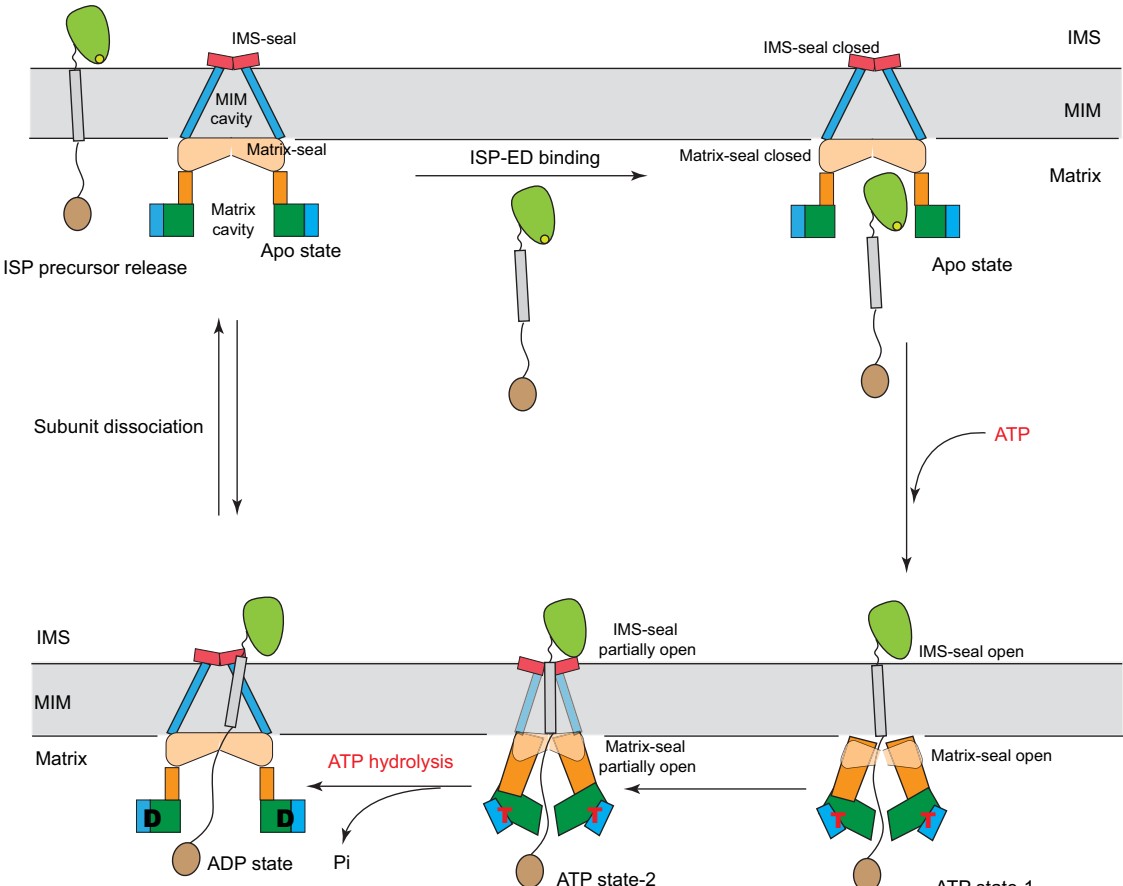

**Fig. 6 | Proposed model for ISP precursor translocation by Bcs1L.** Three nucleotide-binding states of mBcs1L are depicted schematically: Apo, ATP, and ADP. The ATP state is represented by two conformations: ATP state-1 and ATP state-2. Bound ATP is shown as T and ADP as D. The TM domain is shown in blue, Bcs1-specific domain in orange and AAA domain in green for the large domain and blue for the small domain. The mitochondrial inner membrane (MIM) is represented by two parallel horizontal lines and shaded gray. The substrate ISP precursor is also shown. The two substrate cavities were shown and labeled as Matrix cavity and MIM cavity, respectively. Also depicted are two seals: the IMS-seal and the Matrix-seal. In the apo/ADP conformation, both IMS-seal and Matrix-seal are closed. In the ATP state-1 conformation, both Matrix-seal and IMS-seal are open transiently, allowing passage of ISP-ED. In the ATP state-2 conformation, both Matrix-seal and IMS-seal are retracted but not fully closed, wrapping around the TM helix. The ATP hydrolysis resets the mBcs1 to the ADP conformation, presumably allowing the escape of the TM helix. The nucleotide exchange also permits transient release of ISP precursor.

was slowly added to a final concentration of 0.5% to solubilize the membrane. After stirring on ice for 60 min, the solubilized membrane was subjected to ultracentrifugation at 125,000 x g for 30 min to remove insoluble materials. The supernatant was subsequently mixed with Ni-NTA resin (Qiagen) pre-equilibrated in buffer A (25 mM Tris, pH 7.5, 300 mM NaCl, 10% glycerol, 0.05% DDM (Dodecyl-D-Maltoside, Anatrace)) supplemented with 25 mM imidazole at 4 °C for 2 h. The slurry was then packed into a gravity flow column and washed with buffer A supplemented with 75 mM imidazole for 10 column volumes, and the bound mBcs1L was eluted from the resin using buffer A supplemented with 250 mM imidazole. The eluted sample was concentrated using an Amicon Ultra-15 (MWCO 100 kDa) and further purified using a Superdex™ 200 Increase 10/300 GL column (Cytiva) pre-equilibrated with SEC buffer (20 mM Tris, pH 8, 200 mM NaCl, 0.05% DDM) at the flow rate of 0.5 ml/min.

### Isolation of the ISP-ED from bovine cyt bc₁ complex

Purification of cyt $bc_1$ complex from the bovine heart mitochondria follows the established protocol[58]. The isolation of the ISP-ED from the bovine mitochondrial cyt $bc_1$ using limited proteolysis was performed following the procedure previously described[43]. Briefly, cyt $bc_1$ complex at 1 mg/mL in 100 mM NaCl, 5 mM CaCl₂, 0.2% potassium deoxycholate, 0.05% Triton X-100, 20 mM K⁺/MOPS pH 7.2 was mixed with Thermolysin (Promega Corporation, Fitchburg, WI) at a mass ratio of

50:1. After 4 h of incubation at room temperature, the digestion reaction was stopped by adding EDTA to a final concentration of 5 mM. The digested cyt $bc_1$ was dialyzed against a buffer containing 10 mM K⁺/MOPS pH 7.2, 100 mM NaCl, 5 mM EDTA using a 10 kDa MWCO dialysis cassette (Fisher Scientific, Rockford, IL) overnight followed by ultracentrifugation at 150,000 x g, 4 °C for 1 h to remove all hydrophobic materials including undigested cyt $bc_1$. The straw-colored supernatant contains the ISP-ED and was analyzed by SDS-PAGE (Supplementary Fig. 3B). The molecular weight of the digestion product was further confirmed by LC-MS analysis (Supplementary Fig. 3C), which matched the theoretical mass of the cleaved ISP-ED without the $Fe_2S_2$ cluster.

### Mass spectrometry analysis

The intact protein mass was measured on a X500B Q-TOF (Sciex) mass spectrometer coupled to an Exion UHPLC. Protein samples were acidified with 1% formic acid immediately before injection onto an Aeris 3.6 widepore XB-C8 column ($n = 1$ sample for each experiment). Data were analyzed using Explorer and mass reconstruction was performed in BioToolKit, both within the SCIEX OS software.

### ATPase activity

ATPase activity of mBcs1L in the presence or absence of ISP-ED was determined following the previously published protocol[38]. Briefly, the assay was performed in a total of 50 µL reaction mix containing 50 mM

Tris-HCl pH 8.0, 4 mM ATP, 20 mM MgCl$_2$, 1 mM EDTA, and 2–5 µg of protein. After 15 min incubation at 37 °C, the reaction was stopped by adding 800 µL of dye buffer that contains freshly mixed 0.045% malachite green and 1.4% ammonium molybdate tetrahydrate in 4 N HCl in 1:3 ratios. After 1 min, 100 µl of 34% sodium citrate solution was added, and 16 µl of 10% Tween-20 was added 10 min later. Absorbance was measured at 660 nm and compared to the standard curve established by a known amount of KH$_2$PO$_4$ dissolved in the assay buffer, from which the amount of inorganic phosphate released is calculated.

## Pull-down experiment

Purified ISP-ED ( ~ 0.2 mg/ml, 150 µL) was diluted with 800 µL of binding buffer (mBcs1L SEC buffer supplemented with 10 mM of imidazole). The diluted ISP-ED solution was split equally into two halves; to one half was added another 100 µL of binding buffer and 100 µL of pre-equilibrated Ni-NTA resin (QIAGEN), and the other half was mixed with 100 µL of hexahistidine-tagged mBcs1L (1.4 mg/ml) and 100 µL of pre-equilibrated Ni-NTA resin. After leaving on a rotator at 4 °C for 2 h, each protein-resin mixture was transferred to a Bio-spin column (Cat # 732-6008, Bio-Rad) and the flow-through was collected. The resin in the spin column was washed twice with 1 mL of binding buffer, named Wash 1 and Wash 2, respectively. Then, 100 µL of elution buffer (SEC buffer supplemented with 300 mM of imidazole) was used to elute the protein bound to the Ni-NTA resin and the sample was collected as Elution 1. It was repeated as Elution 2. The samples collected throughout the pulldown assay were analyzed on a 12-well, 12% Bis-Tris SDS-PAGE gel and western blot. The ISP-ED was detected by anti-Rieske bundle (Santa Cruz Biotechnology, sc-529220 with 1:1000 dilution for the primary antibody and 1:10,000 dilution for the secondary antibody) and mBcs1L was detected by HisProb-HRP (Thermo Fisher Scientific, PI15165, 1:5,000 dilution).

## Cryo-EM grid preparation and data collection

For the mBcs1L alone sample, 200-Mesh R1.2/1.3 copper grids with a layer of 2 nm continuous carbon film (Quantifoil) were glow-discharged for 60 s with a current of 15 mA using a PELCO easiGlow system. The cryo-grid preparation was done using a FEI Vitrobot Mark IV (ThermoFisher Scientific), which was kept at 4 °C and 95% humidity. ATP and MgCl$_2$ were added to a final concentration of 2 mM and 20 mM, respectively, immediately before applying sample onto the grid. 3 µL of mBcs1L (0.2 mg/ml, or 0.6 µM) with ATP/MgCl$_2$ were applied onto the grid and blotted for 3 s with a blot force of 20 before being plunge-frozen in liquid ethane. The sample was imaged on a Titan Krios TEM (ThermoFisher Scientific) operated at 300 kV at the Center for Molecular Microscopy, NCI. SerialEM (v3.8) was used for automated data collection. Micrographs were recorded on a Gatan K2 Summit (Gatan, Inc.) in counting mode at a nominal magnification of 29,000, which corresponds to a pixel size of 0.858 Å per pixel with a defocus range of -1 to -2.5 µm. A total of 6,912 movies were collected as 40-frame movie series with a dose rate of ~9.98 e$^-$ pixel$^{-1}$ s$^{-1}$ and a total exposure time of 3.7 s, giving an accumulated dose of ~50 e$^-$ Å$^{-2}$.

For the second mBcs1L alone sample, 300-Mesh R0.6/1 UltrAuFoil gold grids (Quantifoil) were glow-discharged for 60 s with a current of 15 mA using a PELCO easiGlow system before being fabricated according to Meyerson's protocol[59]. Purified mBcs1L (1.5 mg/mL) were mixed with 2 mM ATP and 20 mM MgCl$_2$ on ice for about 10 s before being applied to the grid. The grid was blotted for 3 s with a blot force of 20 at 4 °C and 95% humidity and vitrified in liquid ethane using a FEI Vitrobot Mark IV (Thermo Scientific). Images were collected on a Titan Krios TEM operated at 300 kV equipped with a Gatan Bioquantum K3 direct electron detector (Gatan) and a 20 eV energy slit at the National Cancer Institute/NIH Intramural Research Program Cryo-EM facility, Bethesda MD. SerialEM (v3.8) was used for automated data collection. Data were collected as 50-frame movie series, at a nominal magnification of 105,000 in super-resolution mode (0.415 Å pixel$^{-1}$) and a

defocus range of -0.7 to -2.0 µm, with a total exposure dose of 54.5 e$^-$ Å$^{-2}$ from 2.5 s exposure.

The grid preparation and data collection for the Bcs1/ISP-ED sample is the same except that isolated ISP-ED at a concentration of ~0.3 mg/ml were incubated with mBcs1L (1.5 mg/mL) at a molar ratio of 1:1 (ISP:Bcs1 ring) in the presence of 2 mM ATP and 20 mM MgCl$_2$ on ice for about 30 min before being applied to the UltrAuFoil grid.

## Cryo-EM data processing and model building

Data processing for the ATP state-1 mBcs1L was performed in cisTEM[60]. Gain-correction, movie alignment, CTF estimation, automatic particle picking, and extraction were performed on raw micrographs. Two rounds of 2D classification were performed to remove junk particles and dual heptamer rings. Selected particles from 2D classes were used for ab initio model generation and subsequent refinement without symmetry application. Once a homogeneous model was obtained, 50 cycles of 3D classification with 4 classes were performed, which led to 3 classes resembling the ATPγS state and 1 class resembling the ADP state. After auto-refine, 2 of the 3 ATP state classes were found to contain dual heptamers and thus discarded. The best class of the ATP state particles was refined to 3.77 Å resolution with C1 symmetry, or to 3.13 Å with applied C7 symmetry. The remaining 1 class with resemblance to the ADP state of mBcs1L was refined using C7 symmetry to 9.55 Å. For the refinements performed in cisTEM, the FSC curves were calculated using a generous spherical mask with subsequent solvent correction inside the mask[60,61].

The remaining datasets were processed in cryoSPARC[62]. Movies were aligned and gain corrected before performing patch motion correction and patch CTF estimation. Super-resolution movies were Fourier-binned by 2 during movie alignment. Auto-picked particles were cleaned up through two rounds of 2D classification, and particles in the selected 2D classes were subject to ab initio reconstruction with C1 symmetry. Up to 4 ab initio maps were generated to identify junk particles and to probe for conformational heterogeneity. Heterogeneous refinement and 3D classification were used to sort particles into different conformations. Non-uniform refinement was used to refine the final set of particles for each class. For refinements performed in cryoSPARC, FSC curves were calculated using a tight mask with correction by noise substitution.

To build the ATP state model, the mBcs1L-ATPγS model (PDB:6UKS) was manually docked into mBcs1L ATP state EM map using UCSF Chimera[63], and refined using PHENIX real-space refinement[64]. The apo mBcs1L model (PDB:6UKP) served as the starting model for the ADP state map. For all models, de novo model building was carried out manually in COOT[65] for the N-terminal and TM helices and model adjustment to the mBcs1L-specific region and AAA region together with additions of nucleotides and magnesium atoms. The final models were subjected to refinement and minimization in real space using the PHENIX real-space refinement module[64]. The statistics of model refinement are shown in Table 1. Structure validations were performed with Molprobity[66]. Structural figures were prepared using ChimeraX[67] and PyMOL (The PyMOL Molecular Graphics System, Schrödinger).

## Reporting summary

Further information on research design is available in the Nature Portfolio Reporting Summary linked to this article.

## Data availability

The Cryo-EM density maps and atomic models generated in this study have been deposited in the Electron Microscopy Data (EMD) Bank and in Protein Data Bank (PDB) under accession codes EMDB 41276 and PDB 8TI0 (mBcs1L-ATP state-1 in C1); EMDB 41061 and PDB 8T5U (mBcs1L-ATP state-1 in C7); EMDB 41476 and PDB 8TPL (mBcs1L-ATP state-2 in C1); EMDB 41462 and PDB 8TP1 (mBcs1L-ATP state-2 in C7); EMDB 41095 and PDB 8T7U (mBcs1L-ADP state in C1); EMDB 40954

and PDB 8T14 (mBcs1L-ADP state in C7); EMDB 41609 (mBcs1L bound with ISP-ED), and EMDB 41148 and PDB 8TBY (mBcs1L-Apo state in C1). Source data are provided with this paper.

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

## Acknowledgements

This research was supported by the intramural Research Program of the Center for Cancer Research (CCR), National Cancer Institute (NCI), National Institutes of Health (NIH). All DNA sequencing services were conducted at the CCR Genomics Core, NCI and computation for the EM image reconstruction was carried out using the Biowulf Linux cluster (biowulf.hpc.gov) at the NIH high performance computing center, Bethesda, MD 20892. The Cryo-EM datasets were collected at the Center for Molecular Microscopy, Fredrick National Laboratory for Cancer Research, Fredrick MD, USA and the National Cancer Institute/NIH IRP Cryo-EM facility, Bethesda MD, USA. We thank Dr. Ziqiu Wang at the Center for Molecular Microscopy for assistance with EM data acquisition, Dr. Lothar Esser for insightful discussion, and George Leiman for editorial assistance during the preparation of this manuscript.

## Author contributions

J.Z. performed the work and wrote the paper; A.Z. collected the EM data sets; R.H. collected data and assisted with writing the paper; W.K.T. assisted with experiments; L.M.J. performed Mass-spectrometry; D.X. conceived the project, secured funding, performed experiments, and wrote the paper.

## Funding

## Competing interests

The authors declare no competing interests.
