## [Peer Review File · Nature Communications]

Conformations of Bcs1L Undergoing ATP Hydrolysis Suggest a Concerted Translocation Mechanism for Folded Iron-Sulfur Protein Substrate

Editorial Note: Parts of this Peer Review File have been redacted as indicated to remove third party material where no permission to publish were obtained.REVIEWER COMMENTS

Reviewer #1 (Remarks to the Author):

This is an interesting structural biology study building on the last author Prof. Xia' own previous work and that of Prof. Beckmann, both published in 2020. Those two landmark studies showed, surprisingly, that the Bcs1/Bcs1l mitochondrial inner membrane AAA family translocases form a heptameric (rather than hexameric, as typical of AAA proteins) tightly closed pores. These pores have an inner cavity predicted to be large-enough to accommodate the fully folded RISP/UQCRCFS1 protein, a subunit of electron transport chain complex III. However, in the earlier papers RISP was not included in the cryo-EM preparations. In the current paper, Zhan et al. have performed cryo-EM from preparations of purified mouse Bcs1l with different ATP/ADP ratios and with or without bovine RISP domain containing the electron-transferring Fe-S cluster (named ISP-ED in the manuscript). This is the main novelty of the work.

The results suggest that the Bcs1l heptamer has just two major conformations: ATP- and ADP-bound, with the ISP-ED capture into the cavity taking place in the ADP-bound state and the extrusion of ISP-ED across the membrane into the intermembrane space (IMS) during simultaneous (concerted), forceful hydrolysis of ATP by all subunits. Zhan et al. also shed further light on the N-terminal structures facing the IMS. They suggest that these tails form an IMS seal that opens and closes like an iris shutter during ISP-ED translocation. The concerted ATP hydrolysis mode is novel for AAA family ATPases, but not all that surprising in the case of Bcs1l, which has to translocate a bulky folded protein across the MIM swiftly, without compromising the membrane potential. As also alluded by the authors in the last sentence of the abstract, the paper does not really give insight about how the full, untrimmed RISP protein is translocated in vivo. Such insight may be impossible to achieve using cryo-EM because the extrusion step must be very fast and intermediates difficult to catch.

The manuscript is succinct and well-written. My expertise is not in cryo-EM, so I hope the other reviewers will comment on the technical aspects of the methodology and data. My main concern is that despite the novelty described above, the manuscript in its current form is very structural biological and somewhat lacks wider interest. It would have made the paper much more interesting and useful to non-structural biologists if the authors had, given their expertise and resources in the field, selected at least one homozygous patient missense mutation and performed similar structural analysis with it, potentially being able to assess the effect on ISP-ED docking and/or the conformational changes upon ATP hydrolysis and the rest of the translocation cycle.

Additionally, the authors should discuss what implications their further structural insight (as compared to their 2020 paper) has in relation to the numerous patient mutations in BCS1L, causing mitochondrial disease in humans.

Reviewer #2 (Remarks to the Author):

In this study, Xia and coworkers use cryo-EM to investigate the conformational states and nucleotide occupancies of the AAA+ protein translocase Bcs1L during ATP hydrolysis and in the absence or presence of its ISP substrate. The Bcs1L-mediated translocation of folded, Fe₂S₂-containing ISP across the inner mitochondrial membrane is a critical step in cytochrome bc₁ complex assembly. Previous work by the authors and others solved the structures of Bcs1 bound to ATPgS or ADP, and indicated that the heptameric complex uses concerted ATP hydrolysis and conformational changes to modulate its two internal compartments and push the functional domain of ISP across the membrane in an airlock-like mechanism.

Here, the authors present the cryo-EM structures of mouse mBcs1L with all seven subunits uniformly occupied by ATP or no nucleotide, as well as an all-ADP-bound state with density for ISP in the matrix cavity. The ATP-bound structure is highly similar to the prior structure of Bcs1 in complex with the slowly hydrolyzed ATPgS, showing no indication for a spiral arrangement of subunits or the presence of different nucleotides in the heptamer.

However, the presented study is overall very incremental and largely confirmatory, not only compared to the previous structural work that already proposed and well supported a concerted ATP-hydrolysis and conformational change model, but also with respect to a high-speed AFM-based study that validated a concerted hydrolysis mechanism, proposed highly similar models as presented here, was co-authored by Xia, and just published last month in Nature Communications (Pan et al., NatComm 14). The present manuscript would have been a good addition to this AFM story, but on its own certainly does not provide a significant advance that justifies consideration for publication in Nature Communications.

In the end, this study primarily addresses whether the presence of ISP affects or changes the previously proposed concerted mechanism of Bcs1, which is not the case. Also, the density observed for ISP in the matrix cavity of ADP-bound mBcs1L is too low in resolution to draw any conclusion.

The only new aspect is that the authors were able to resolve a small helix at the N-terminus of mBcs1L (~12 additional amino acids compared to the Bcs1 structures previously solved by the Beckman and Xia groups), yet the presented interpretations regarding different angles between this small helix and the transmembrane region or this helix lying flat on the IMS side of the mitochondrial inner membrane are questionable in the absence of a membrane.

Furthermore, the authors describe two different ATP-bound states, yet their comparison and functional interpretations are egregious. These reconstructions were obtained from two different data sets that came from different grid types, had a 30-fold difference in the number of particles (997,041 versus 33,415), and were analyzed with different software (cisTEM versus CryoSparc). Those data sets also apparently showed different fractions of tetradecamer species, and it is unclear whether they were acquired with the same or different mBcs1L protein preparations. Drawing any mechanistic conclusions from observed structural or conformational differences is therefore not appropriate.

In summary, as stated above, this manuscript does not provide sufficiently new insight and is therefore not suited for publication in Nature Communications.

Reviewer #3 (Remarks to the Author):

The manuscript by Zhan et al. builds upon a previous manuscript by the same group (Tang et al. NSMB, 2020) that reported structures of mouse Bcs1, a mitochondrial membrane-bound AAA protein that translocates folded Rieske iron-sulfur protein (ISP) across the mitochondrial inner membrane. Two different conformational states were obtained via X-ray crystallography and cryoEM, thereby providing a potential mechanism for the translocation of folded proteins across the membrane by Bcs1. However, how ISP is bound and released by Bcs1, and whether ATP hydrolysis follows a sequential pattern as seen in many other AAA ATPases, or rather via a concerted mechanism, is currently unclear.

The current work by Zhan et al. presents cryoEM structures of mouse Bcs1L during active ATP hydrolysis in the absence and presence of substrate (bovine ISP). The cryoEM structures reveal that while multiple conformations are present within the collected datasets, none of these show co-existing nucleotide states (apo/ADP/ATP) within the heptamer, and all conformations are devoid of a characteristic 'seam' observed in AAA ATPases that follow a sequential ATP hydrolysis mechanism. Based on these structures, the authors conclude that Bcs1 subunits act in a concerted manner to translocate substrates. Furthermore, a structure is obtained of mouse Bcs1L in complex with bovine ISP-ED in the ADP-bound form, which shows the substrate trapped in the proposed substrate-binding cavity of Bcs1L. Finally, a cryoEM map of apo mBcs1L reconstructed without imposing symmetry shows an asymmetric rearrangement of individual subunits and a large crevice between some subunits, potentially corresponding to a state where ISP precursor can be released.

The study by Zhan et al. provides valuable advances on the translocation mechanism of ISP by Bcs1L that will undoubtedly be of interest to the field, and I therefore support publication in Nature Communications. While the cryoEM work seems to be carefully performed, I do have a few technical questions and issues that I believe need to be addressed, as well as some general comments on the manuscript itself.

Here are my comments and remarks:

Introduction:

- It is unclear to me why the study by Pan et al. (Nature Communications, October 11th, 2023), which includes both the first and last author of this manuscript, and which shows a concerted mechanism for mouse Bcs1L via high-speed atomic force microscopy (AFM), is not referenced. The study by Pan et al. is highly complementary to the cryoEM study presented here, and referencing it in the manuscript would strengthen the main observations and conclusions made.

- In the Introduction the authors refer to a set of papers (references 3-16) and mention a "universal threading model" that follows sequential ATP hydrolysis. However, reference 8 (Jessop et al. Communications Biology, 2020) describes an AAA ATPase structure with a double seam conformation, similar to structures observed for ClpX (Glynn et al, Cell, 2009, Stinson et al, Cell, 2013) for which a stochastic/probabilistic rather than sequential ATP hydrolysis mechanism is proposed. The authors refer to the paper by Gatsogiannis et al., 2019 (ref. 16) which reports the cryoEM structure of ClpXP, but

should also refer to the ClpXP structure papers by Ripstein et al. Elife, 2020 and Fei et al. Elife, 2020 which propose a sequential and stochastic ATP hydrolysis mechanism respectively (see also commentary by Tsai & Hill, Elife, 2020). I believe the possibility of stochastic ATP hydrolysis for some AAA ATPases needs to be briefly discussed in the introduction.

- Line 146: 'Using established method (35), ...' should be changed to 'Using an established method (35), ...'.

Results

- For the preparation of cryoEM grids of mBcs1L with added ATP and MgCl₂, the reaction is allowed to proceed for 10 s before plunging, using a concentration of 0.2 mg/ml (0.6 μM) of mBcs1L, 2 mM ATP and 20 mM MgCl₂. For the sample containing both ATP and ISP-ED substrate however, the reaction is allowed to proceed for 30 min, using concentrations of 1.5 mg/ml (4.5 μM) of mBcs1L, 2 mM ATP and 20 mM MgCl₂. Given the turnover rate of just below 1 ATP/heptamer/second (42 ATP/heptamer/min), is mBcs1L still undergoing active ATP hydrolysis after 30 min under these conditions? Could this be tested in an ATPase activity assay such as described on p.7 of the manuscript?

- Pull-down experiments were conducted to show a direct interaction between mouse Bcs1L and bovine ISP-ED. Is anything known about the affinity between Bcs1 and ISP? Were any biophysical methods (BLI/SPR,...) tried to obtain kinetic affinity and association/dissociation rates?

- On p.8 it is stated that “without exception all tested AAA proteins were captured with co-existing different nucleotide states (Apo/ADP/ATP) in protomers that arrange in spiral conformations (3 – 16).” This is incorrect for reference 8 where a double-seam state is observed in addition to the spiral (one-seam) conformation (see second comment on Introduction).

- ISP-ED seems to be present in the Class 3 map, and the interaction appears to be mediated via one or two of the seven equivalent subunits. Could it be tried to perform a focused classification and/or local refinement using a mask around the density allocated to ISP-ED and the two subunits it seems to be interacting with, in an attempt to increase the quality of the map at that region? The molecular weight of two mBcs1L subunits bound to ISP-ED is $(47.7 \times 2) + 14.4 = 109.8$ kDa which might be large enough to try focused classification and/or local refinement.

- On p.15 (paragraph “Apo Bcs1L displays asymmetrical subunit arrangement) the section should start by mentioning that an additional dataset was collected in the absence of both substrate and nucleotides. If I am not mistaken, this is not yet mentioned at any point in the manuscript when starting the respective section.

- Extended Data Figure 7A contains a typo: the amount of particles for ATP state-2 is 13,851, not 213,851.

-Extended data Fig. 10 shows a modeled PE lipid, but no fit is shown in the experimental map. I think its necessary to show the actual fit in the map as is done for ATP molecules in Figure 2.

Discussion:

- The discussion would benefit from a brief mention of the possibility of stochastic rather than sequential ATP hydrolysis for some AAA ATPase family members (see second comment on Introduction).

- On p.22, the sentence “Recent and current structural studies have offered important insights into how Bcs1 meets these requirements and are in support of the concerted translocation mechanism” is missing references.

Methods:

- For all of the processed cryoEM datasets and structural refinements, map-to-model FSC curves are absent. Map-to-model FSC curves should be calculated using unsharpened and unfiltered maps, and the FSC0.5 values should be reported in table 1 for each dataset.

- In Table 1 it is not stated which B-factors were used for sharpening of the maps.

- FSC curves shown in the workflow figures do not state what mask is used to calculate the FSC curves. For the refinements performed in cryoSPARC I assume the shown FSC curves are FSC curves calculated using a tight mask with correction by noise substitution? This should be explicitly mentioned.

- Line 641: 'Ab Initial model generation' should be replaced by 'Ab Initio model generation'.

- Line 645: 'The best class of the ATP state particles were refined' should be changed to 'The best class of the ATP state particles was refined'

- Line 649: 'The rest datasets' should be changed to 'The rest of the datasets' or 'The remaining datasets'.

Point-by-point response to reviewers' comments

Reviewer #1

Reviewer 1's comment: My main concern is that despite the novelty described above, the manuscript in its current form is very structural biological and somewhat lacks wider interest. It would have made the paper much more interesting and useful to non-structural biologists if the authors had, given their expertise and resources in the field, selected at least one homozygous patient missense mutation and performed similar structural analysis with it, potentially being able to assess the effect on ISP-ED docking and/or the conformational changes upon ATP hydrolysis and the rest of the translocation cycle.

Additionally, the authors should discuss what implications their further structural insight (as compared to their 2020 paper) has in relation to the numerous patient mutations in BCS1L, causing mitochondrial disease in humans.

Authors' response: We appreciate the reviewer's comments and suggestions in relation to mutations that lead to mitochondrial diseases, which will motivate us to speed up this line of work. Indeed, this is an ongoing project involving not only expressing mutant proteins for structural and biochemical characterizations but also conducting functional studies using cell biology approach to check phenotypes. Because of the autosomal recessive nature, BCS1L mutations from patients' two defective copies of the gene often have non identical mutations. In our laboratory, we can only evaluate mutant protein containing a single mutation, which may or may not represent accurately the phenotype manifested in patients. Currently, various mutants of BCS1L have been produced bearing a number of mutations identified in GRACE and Bjornstad diseases, including S78G, R155P, R183H, and R183C. While appreciating the reviewer's suggestions, we feel this body of experiments form an integral part of its own, is not likely to be completed within the given period for the revision of this manuscript. Therefore, inclusion of these works in this paper would be premature and beyond the scope of this paper.

We noticed, however, that among the six mutations associated with the GRACILE syndrome (GS)(Tang et al., Nat Str Mol Biol, 2020), which is the severe form of Bcs1L-linked mitochondrial disease, three are arginine mutations R45C, R73C, and R144Q. When these mutations are mapped to the structures of both ADP- and ATP-bound forms of Bcs1L, they clustered together, in both ATP and ADP bound forms, at the boundary between the TM and Bcs1-specific regions and at the subunit interface (See figures below). Interestingly, the other three GS-associated mutations T50A, S78G, and P99L are located to the same cluster. Although these mutations are by no means lethal to the Bcs1 function, their clustering suggests an important role of this region for the proper function of Bcs1L. These observations certainly set the stage for future more detailed studies of Bcs1 function. In the revision, we added the following in the Discussion section.

The need to maintain the membrane integrity is underscored by the observation that all six mutations (R45C, T50A, R73, S78G, P99L, and R144Q) identified from patients with GRACILE syndrome (37), a severe form of Bcs1L-linked mitochondrial disease, are clustered at the border between the TM region and Bcs1-specific region and at the subunit interface (Supplementary Fig. 11).

Supplementary Figure 11. Clustering of mutations associated with GRACILE syndrome. The structures of mBcs1L in different nucleotide-bound forms are rendered in cartoon form overlaid with an electrostatic potential surface, which has the positive potential in blue, negative potential in red, and neutral surface colored white. The mitochondrial inner membrane is demarcated with two horizontal black lines. Definitions for various structural regions of the molecule are also given. The insets on the right represent a small part of the Bcs1-specific region that borders the mitochondrial inner membrane (MIM) and at the interface between two subunits. The six GS-linked mutations are rendered as stick models and labeled. (A) Bcs1L in the ADP-bound form and (B) Bcs1L in the ATP-bound form.

Reviewer #2

Reviewer 2's comment: Here, the authors present the cryo-EM structures of mouse mBcs1L with all seven subunits uniformly occupied by ATP or no nucleotide, as well as an all-ADP-bound state with density for ISP in the matrix cavity. The ATP-bound structure is highly similar to the prior structure of Bcs1 in complex with the slowly hydrolyzed ATPgS, showing no indication for a spiral arrangement of subunits or the presence of different nucleotides in the heptamer.

Authors' response: We thank the reviewer for reading and commenting on our paper. In this work, we presented four structures at high resolution: two ATP bound structures with slightly different conformations, one ADP bound structure, and one apo structure with no applied C7 symmetry, which showed asymmetrical arrangement of AAA domains.

Reviewer 2's comment: However, the presented study is overall very incremental and largely confirmatory, not only compared to the previous structural work that already proposed and well supported a concerted ATP-hydrolysis and conformational change model, but also with respect to a high-speed AFM-based study that validated a concerted hydrolysis mechanism, proposed highly similar models as presented here, was co-authored by Xia, and just published last month in Nature Communications (Pan et al., NatComm 14). The present manuscript would have been a good addition to this AFM story, but on its own certainly does not provide a significant advance that justifies consideration for publication in Nature Communications.

Authors' response: The reviewer thinks of our work being incremental and largely confirmatory, which we respectfully disagree. (1) We tried to combine results of EM and AFM experiments, but we were not happy with the resulting long paper that lacks the coherence stemming from two different approaches and uses of different jargons. (2) Previous works (Tang et al., NSMB, 2020 and Kater et al., NSMB, 2020) were not designed for real-time visualization and time-resolved experiments and focused on structure determinations of Bcs1 in different conformations. Therefore, these works, while being landmark achievements, should not be considered as evidence for the concerted mechanism of Bcs1 function. (3) AFM provides a real time view of conformational change in the presence of ATP with a temporal resolution of 270 μ s. AFM does not tell what type of nucleotides binds to individual Bcs1 subunits. Our EM work used a time-resolved approach to trap reaction intermediates that exist in significant populations during active ATP hydrolysis cycles. EM can identify apo, ADP, or ATP binding at the nucleotide-binding sites by resolving structures at high resolution. The two methods are complementary to each other, not overlapping. The AFM method provides validation for the concerted mechanism up to 270 μ s, whereas the time-resolved approach offers conformational information of intermediates on the instant they were trapped. Together, both methods give a more complete presentation of the concerted mechanism. (4) In this paper, we provide evidence for the first time that Bcs1 also functions in a concerted manner in the presence of substrate. To our knowledge, this is also the first time that concerted translocation mechanism being structurally characterized for AAA proteins.

Reviewer 2's comment: In the end, this study primarily addresses whether the presence of ISP affects or changes the previously proposed concerted mechanism of Bcs1, which is not the case. Also, the density observed for ISP in the matrix cavity of ADP-bound mBcs1L is too low in resolution to draw any conclusion.

Authors' response: Subunits of canonical AAA proteins such as ClpA and p97 often display planer arrangement in the absence of substrates. These proteins adopt a spiral staircase arrangement of their subunits when substrates are trapped in their translocation pores. Therefore, the planer configuration of Bcs1 in the absence of substrate, although consistent with the concerted mechanism, is considered insufficient as a proof. In this work, we further show that this planer configuration persists even in the presence of substrate, contrasting to the commonly observed non-planar arrangement of canonical AAA proteins and offering strong evidence in supporting the concerted mechanism.

With respect to the ISP-ED binding, our current model is based on a 7.2 Å map with 8,300 particles. During the revision, we attempted using local refinement that focused on either 2 Bcs1 subunits and the ISP or the entire AAA domain and the ISP, and resolution of the resulting EM density in the matrix cavity remained the same. The limit in resolution appears to be due to insufficient number of particles but it is not possible to exclude a contribution from low binding affinity. Nonetheless, we see that the shape of the density in the cavity is unchanged with different focusing masks, with good density for the helical regions of ISP-ED (η_1 , α_1 to η_2 , and η_3 , as defined in Supplementary Figure 1B), and relatively weaker density for the β -strands and connecting loops, which is expected at the current resolution. We also tested fitting of ISP model to the density at two different map resolutions, which indicated best fit at about 7 Å resolution (Table below), by the log-likelihood gain measurement (LLG). Additionally, with the 7 Å density map, the ISP-ED could be docked into the density consistently in the same orientation, which indicates a likely correct solution (Figure 3). The reviewer questioned about what conclusion one might draw from this analysis. Although our EM analysis is not at high resolution, it is consistent with our pull-down assay and the binding orientation of ISP-ED can be established. It is also established that the binding is skewed involving only 2 subunits in the heptameric ring.

Table. Fitting ISP-ED model into EM density.*

Map resolution (Å)	8	7.2
Map LLG	81.4	116.1
Map CC	0.68	0.64

* - Phenix was used for model fitting.

Reviewer 2's comment: The only new aspect is that the authors were able to resolve a small helix at the N-terminus of mBcs1L (~ 12 additional amino acids compared to the Bcs1 structures previously solved by the Beckman and Xia groups), yet the

presented interpretations regarding different angles between this small helix and the transmembrane region or this helix lying flat on the IMS side of the mitochondrial inner membrane are questionable in the absence of a membrane.

Authors' response: In the following, we list novel findings of our work, which includes: (1) Using a time-resolved approach coupled to cryo-EM analysis, we provide evidence for a concerted mechanism both in the absence and presence of substrate, which differs from the sequential or stochastic mechanisms described for canonical AAA proteins. (2) We provide the complete structures of mouse Bcs1 in both the ADP state and ATP state 2 from residues 1-418, leading to the structural description of the IMS seal. (3) Our work also showed for the first time the ordered TM helices in the ATP-bound form, permitting analysis of conformational change of the TM helices between ATP and ADP states. This analysis reveals that the IMS seal undergoes an iris like opening and closing motion when transitioning between ATP and ADP states. (4) Our structure allows visualization of the binding of substrate ISP-ED to Bcs1 in a pre-translocation ADP-bound state, which is consistent with our pull-down assay demonstrating the binding of ISP-ED to Bcs1. (5) We also showed the relatively weak and asymmetric arrangement of Bcs1 subunits in apo state by cryo-EM analysis, by truncation mutagenesis coupled to Blue-Native PAGE, and by mass spectrometry, which indicates a plausible release mechanism for ISP where the apo Bcs1 ring opens up laterally to allow the bound ISP to diffuse out post translocation.

With respect to the paragraph containing "In the ADP state, the first 11 residues at the N-terminus (residues M1 to K11) form a 3-turn short helix lying flat on the IMS side of the MIM ...", our description was not accurate. In the revision, this paragraph is revised to "In the ADP state, the first 11 residues at the N-terminus (residues M1 to K11) form a 3-turn short helix, which lies parallel to the surface of MIM and is connected by a short loop to the 50 Å long, 9-turn TM helix (residues P14 to Y47) that spans the MIM (Fig. 5A, left). This N-terminal assembly consisting of the 3-turn short helix, the connecting loop, and the beginning (¹⁴PYF¹⁶) of the long TM helix makes contacts with their counterparts of neighboring subunits to form a seal on the IMS side of the membrane, hereafter referring to as IMS-Seal (Fig. 5B, left). The IMS-Seal in the ADP state features a narrow opening of approximately 9 Å in diameter, which is constructed by the side chains of Y15 pointing up and towards the center."

Reviewer 2's comment: Furthermore, the authors describe two different ATP-bound states, yet their comparison and functional interpretations are egregious. These reconstructions were obtained from two different data sets that came from different grid types, had a 30-fold difference in the number of particles (997,041 versus 33,415), and were analyzed with different software (cisTEM versus CryoSparc). Those data sets also apparently showed different fractions of tetradecamer species, and it is unclear whether they were acquired with the same or different mBcs1L protein preparations. Drawing any mechanistic conclusions from observed structural or conformational differences is therefore not appropriate.

Authors' response: We thank the reviewer for the comments on the presence of the two ATP-bound Bcs1 data sets. In the text, we discussed the two data sets. "As mentioned earlier, two

independent ATP states were identified in this work: ATP state-1 and ATP state-2. Their overall conformations resemble the previously reported ATP γ S-bound state (32). The AAA ATPase regions are very much the same, but they differ substantially in the TM region and the Bcs1-specific region when compared to the ATP γ S-state and to each other (Supplementary Table 2 and Supplementary Fig. 6). In the ATP-1 state, the entire IMS-seal and the TM helices are disordered (Figure 5A, middle), which has also been reported for the ATP γ S-state (32). By contrast, the ATP-2 state reveals a near-complete structure of the TM region, showing a similar membrane-embedded basket encircled by TM helices as in the ADP state (Fig. 5A, right)." We want to clarify that the two data sets came from the same batch of protein that was similarly treated. Both data sets were processed also using both cisTEM and CryoSparrc software showing the same results. In the following figure, we include 2D class averages from the four data sets (ATP γ S, ATP-1, ATP-2, ATP-2 with ISP) showing clearly the visible difference in the TM region, which indicate that the difference is not related to the number of particles used for the reconstructions. The ATP-1 state is similar to the ATP γ S state that has been reported in our previous publication (Tang et al., 2020, NSMB, ATP γ S-bound, PDB 6UKS and EMD-20811), featuring a pointing TM region, while the ATP-2 state has been repeatedly seen in the absence or presence of ISP-ED (Supplementary Figures 5 and 7), displaying a more flat TM region. Most importantly, we want to provide readers with complete and unbiased information about the presence of nuanced features in the ATP-bound Bcs1. We should also point out that in the yeast Bcs1 structure (Kater et al., 2020 NSMB), two conformations were described for Apo Bcs1. [REDACTED]

Reviewer #3

Authors' response: We thank the reviewer for his/her positive comments

Reviewer 3's comment: - It is unclear to me why the study by Pan et al. (Nature Communications, October 11th, 2023), which includes both the first and last author of this manuscript, and which shows a concerted mechanism for mouse Bcs1L via high-speed atomic force microscopy (AFM), is not referenced. The study by Pan et al. is highly complementary to the cryoEM study presented here, and referencing it in the manuscript would strengthen the main observations and conclusions made.

Authors' response: We agree with the reviewer. The Nature Communication paper by Pan et al came out after our manuscript was submitted. In the revision, we added the following description along with the reference in the last paragraph of the Introduction section.

"Recently, this issue was approached by the high-speed atomic force microscopy and line scanning (HS-AFM-LS) method in real time and it was found that subunits of mBcs1L hydrolyze ATP in a concerted manner within the detection limit of the method (270 μ s) (40)."

Reviewer 3's comment: - In the Introduction the authors refer to a set of papers (references 3-16) and mention a "universal threading model" that follows sequential ATP hydrolysis. However, reference 8 (Jessop et al. Communications Biology, 2020) describes an AAA ATPase structure with a double seam conformation, similar to structures observed for ClpX (Glynn et al, Cell, 2009, Stinson et al, Cell, 2013) for which a stochastic/probabilistic rather than sequential ATP hydrolysis mechanism is proposed. The authors refer to the paper by Gatsogiannis et al., 2019 (ref. 16) which reports the cryoEM structure of ClpXP, but should also refer to the ClpXP structure papers by Ripstein et al. Elife, 2020 and Fei et al. Elife, 2020 which propose a sequential and stochastic ATP hydrolysis mechanism respectively (see also commentary by Tsai & Hill, Elife, 2020). I believe the possibility of stochastic ATP hydrolysis for some AAA ATPases needs to be briefly discussed in the introduction.

Authors' response: We agree with the reviewer that the models for using a stochastic, instead of sequential, ATP hydrolysis among subunits of AAA proteins cannot be excluded for AAA proteins like RavA and ClpX. In the revision, the following sentence has been revised and corresponding references added in the Introduction section.

" However, the debate over whether AAA proteins carry out ATP hydrolysis via a sequential (3-15) or a probabilistic mechanism remains (16-21)."

In the Discussion section, the following paragraph is also revised.

"Using similar experimental designs as shown in this work, previous cryo-EM studies of

canonical, hexameric AAA ATPases revealed a shared threading mechanism, that is characterized by a dominant spiral conformation (3-15). Five of the six subunits form a descending right-handed spiral around a polypeptide chain or DNA substrate, while the sixth "seam" subunit lies in the middle to bridge the top and bottom of the spiral (48). Except for the "seam" subunit, the five other subunits in the spiral all have their conserved pore-loops protruding into the central pore in direct contact with the substrate. The nucleotide occupancy for each individual subunit is highly correlated with its position in the spiral, with the top three subunits always bound with ATP and the "seam" subunit in the post-hydrolysis state (Apo/ADP). The two subunits at the bottom of the spiral were frequently seen in the post-hydrolysis state. Substrate translocation is thought to go through sequential ATP hydrolysis, as individual subunit transitions from ATP-bound form to extended ADP-bound form, thus moving downwards while pulling the substrate along stepwise. However, this sequential mechanism faces difficulties with the observations from some AAA proteins like RavA and ClpX, which feature two separate seam subunits in cryo-EM structures, leading to the proposed probabilistic mechanism for ATP hydrolysis (16-21). "

Reviewer 3's comment: - Line 146: 'Using established method (35), ...' should be changed to 'Using an established method (35), ...'.

Authors' response: corrected.

Reviewer 3's comment: - For the preparation of cryoEM grids of mBcs1L with added ATP and MgCl₂, the reaction is allowed to proceed for 10 s before plunging, using a concentration of 0.2 mg/ml (0.6 μM) of mBcs1L, 2 mM ATP and 20 mM MgCl₂. For the sample containing both ATP and ISP-ED substrate however, the reaction is allowed to proceed for 30 min, using concentrations of 1.5 mg/ml (4.5 μM) of mBcs1L, 2 mM ATP and 20 mM MgCl₂. Given the turnover rate of just below 1 ATP/heptamer/second (42 ATP/heptamer/min), is mBcs1L still undergoing active ATP hydrolysis after 30 min under these conditions? Could this be tested in an ATPase activity assay such as described on p.7 of the manuscript?

Authors' response: To address the question by the reviewer whether Bcs1 is still active after 30 minutes, we performed the experiment suggested by the reviewer. We incubated Bcs1 (0.3 mg/mL or 0.9 μM) on ice with 2 mM ATP for 30, 40, 50, and 60 min to measure the amount of Pi released. The plot below shows Pi accumulation over time. The gradual increase of [Pi] over the incubation period suggests that Bcs1 was actively hydrolyzing ATP during the 60 min time course. However, the decreased rate suggests either ADP inhibition, protein inactivation or both. As a comparison, in the paper published by Kater et al (2020 NSMB) on the structure of yeast Bcs1/ADP, the sample was prepared by incubating yeast Bcs1 (0.5 mg/mL) with 1 mM ATP for 1 hr.

Reviewer 3's comment: - Pull-down experiments were conducted to show a direct interaction between mouse Bcs1L and bovine ISP-ED. Is anything known about the affinity between Bcs1 and ISP? Were any biophysical methods (BLI/SPR,...) tried to obtain kinetic affinity and association/dissociation rates?

Authors' response: To our knowledge, the K_d measurements between Bcs1 and substrate ISP or that between Bcs1 and ISP-ED have not been reported.

Per reviewer's suggestion, we considered several commonly used methods for measuring or estimating K_d values for ISP-ED binding to mBcs1L, which included ITC, AUC, SPR, BLI, etc. Because Bcs1 is a membrane protein, its TM region is associated with a very large detergent micelle (Supplementary Figures 5, 7 and 8), which makes small sized change difficult to detect. During protein preparation, any attempt to concentrate the protein inevitably leads to changes in the concentration of detergent micelles. Moreover, mBcs1 cannot be concentrated to high concentrations (~ 5 mg/mL). Another potential issue is the fact that because ISP-ED only binds to the matrix cavity of Apo and ADP-bound Bcs1 and it does not invoke conformational or size change in Bcs1, it is conceivably difficult to use surface plasmon resonance (SPR) or bio-layer interferometry (BLI) methods, especially when Bcs1 is used as the immobilized phase.

Nevertheless, we decided to try BLI. As a control, we first tested BLI responses to Bcs1's conformational change induced by ATP γ S, which goes from the 14 nm diameter in mBcs1L's AAA-domain in the ADP-bound form to 13 nm in the ATP γ S-bound form and the overall height of the molecule is reduced by 0.7 nm as illustrated from our EM and AFM studies. The experiment was conducted with biotinylated Bcs1 immobilized on a SA (streptavidin) sensor or a NTA sensor, and a buffer solution containing 4 mM ATP γ S/20 mM Mg $^{2+}$ was used as analyte (see below for the raw data). No significant signal was observed from the Bcs1 immobilized on the sensor, suggesting the conformational changes induced by ATP γ S binding could not be detected by BLI in our experiment.

We went ahead to test ISP-ED binding to mBcs1L anyway. Although using ISP-ED as an immobilized phase would be ideal to conduct the experiment, we had to use mBcs1L as the immobilized phase due to insufficient concentration of mBcs1L. By using ISP-ED at different concentrations as the analyte solution, we found significant non-specific binding between ISP-ED to SA sensor that is proportional to ISP concentration. The non-specific binding could be quenched with buffer containing 50 $\mu\text{g/ml}$ biocytin and 500 mM NaCl but not eliminated. Under such optimized conditions, with 15 mM ISP-ED as analyte solution (the highest concentration we had), the signal difference between experimental sensor and the control sensor is discernable but still very small (see below for the raw data), and the binding did not reach saturation after 180 second association step, indicating the K_d is weak probably in the μM range. Under current conditions, we think it is unlikely that a complete set of curves could be obtained. Curve fitting of this data gave an estimated K_d of $\sim 1 \mu\text{M}$ with vary large errors. The inaccurate measurement could be due to a combination of factors: (1) the binding affinity between ISP-ED and Bcs1 may be weak in the micromolar range; (2) the protein concentrations in the analyte phase is still too low; (3) the change in optical thickness induced by ISP-ED binding to mBcs1L is rather small due to micelle interference; and (4) the non-specific binding between ISP-ED and the sensor interferes with the measurement.

Due to the difficulty in obtaining proteins in higher concentration and in large quantity, and the buffer mismatch issue arising from concentrating membrane proteins, we did not attempt analytic ultracentrifugation (AUC) and isothermal calorimetry (ITC) . We tested the use of differential scanning fluorimetry (DSF) that has been used widely to determine K_d values for

small molecule binding to proteins. The result was ambiguous to interpret due to the fact that the fluorescent dye SYPRO orange interacts with Bcs1, ISP, and their complex.

We also tried making a rough estimate of the K_d value based on the percentage of particles obtained from EM experiment and on the equation ($K_d = \frac{[ISP-ED]_{free} [Bcs1-ADP]_{free}}{[Bcs1-ADP-ISP]}$), where estimated $[ISP-ED]_{free} \sim 3.87 \mu M$, $[Bcs1-ADP]_{free} \sim 2.84 \mu M$, and $[Bcs1-ADP-ISP]_{free} \sim 0.63 \mu M$, which gave an estimated K_d of $17 \mu M$. These numbers were derived from our initial Bcs1 protein concentration at 1.5 mg/ml ($4.5 \mu M$), and initial $[ISP-ED] = 4.5 \mu M$. We assumed that the percentage of our final selected Bcs1 particles (Supplementary Figure 7) is proportional to the initial concentrations:

14% $[Bcs1-ADP-ISP]$: ($4.5 \mu M * 0.14 = 0.63 \mu M$), 63% $[Bcs1-ADP]_{free}$: ($4.5 \mu M * 0.63 = 2.84 \mu M$), and 23% $[Bcs1-ATP]_{free}$: ($4.5 \mu M * 0.23 = 1.04 \mu M$). The final $[ISP-ED]_{free} = 4.5 \mu M - 0.63 = 3.87 \mu M$.

It is noted that this estimation of K_d does not consider the particles classified as "junk particles" during the 2D and 3D classifications, neither does it take into account the amount of particles removed during blot procedure. If we assume 50% particles were removed and keep the percentage among the different classes the same, then the estimated K_d would rise to $19 \mu M$. Perhaps, these estimates probably represent an upper- and a lower-limit for the binding affinity of ISP-ED to the mBcs1. Accurate measurement needs better instrumentation and higher protein concentrations, which will be pursued in our future work.

In the revision, we added " The low percentage of particles with ISP-ED occupied (14% of the total particles used in the reconstruction) could be the result of ISP-ED being constantly translocated by mBcs1L, while actively hydrolyzing ATP or due to relatively low binding affinity." in the section entitled "Extra density the size of ISP-ED found only in the Class 3 map"

Reviewer 3's comment: - On p.8 it is stated that "without exception all tested AAA proteins were captured with co-existing different nucleotide states (Apo/ADP/ATP) in protomers that arrange in spiral conformations (3 – 16)." This is incorrect for reference 8 where a double-seam state is observed in addition to the spiral (one-seam) conformation (see second comment on Introduction).

Authors' response: The sentence "Similar experiments have been performed with various AAA proteins, and without exception all tested AAA proteins were captured with co-existing different nucleotide states (Apo/ADP/ATP) in protomers that arrange in spiral formations (3-16)." was changed to "Similar experiments have been performed with various AAA proteins, and majority of tested AAA proteins, when substrates were present, were captured with co-existing different nucleotide states (Apo/ADP/ATP) in protomers that arrange in non-planar formations (3-20).".

References 17-20 are added.

Reviewer 3's comment: - ISP-ED seems to be present in the Class 3 map, and the interaction appears to be mediated via one or two of the seven equivalent subunits. Could it be tried to perform a focused classification and/or local refinement using a mask around the density allocated to ISP-ED and the two subunits it seems to be interacting with, in an attempt to increase the quality of the map at that region? The molecular weight of two mBcs1L subunits bound to ISP-ED is $(47.7 \times 2) + 14.4 = 109.8$ kDa which might be large enough to try focused classification and/or local refinement.

Authors' response: To separate Bcs1-ADP particles with bound ISP-ED from those without, 3D classification with a focus mask on the ISP-ED and the 4 neighboring AAA domains was performed, which effectively sorted out the two populations and enhanced the ISP-ED density in the matrix cavity of Bcs1 (Class 3 in Supplementary Fig. 7, obtained after the 3D classification with a focus mask and homogenous refinement afterwards).

To further improve the quality of the Bcs1-ISP map, we performed focused refinement with either a mask covering ISP-ED and its two neighboring Bcs1 subunits, or a mask covering only the AAA-domain of Bcs1 ring and the ISP, and in both cases the resolution in the matrix cavity still did not go beyond 7 Å. Nonetheless, we saw the shape of the density in the cavity remains unchanged with different focusing masks, with good density for the helical regions of ISP-ED (η_1 , α_1 to η_2 , and η_3 , which are defined in Supplementary Figure 1B), and relatively poor density for the β -strands and connecting loops, which is expected at the current resolution (illustrated in Figure 3B). With the ~ 7 Å density map, the ISP-ED could be docked in consistently in the same orientation with a LLG of over 100, which indicates a likely correct solution. The docked ISP-ED/Bcs1 model was subject to real-space refinement in Phenix and the statistics have been added to Table 1. To resolve interactions of specific residues in the binding interface, a Bcs1-ISP map at higher resolution is required.

Reviewer 3's comment: - On p.15 (paragraph "Apo Bcs1L displays asymmetrical subunit arrangement) the section should start by mentioning that an additional dataset was collected in the absence of both substrate and nucleotides. If I am not mistaken, this is not yet mentioned at any point in the manuscript when starting the respective section.

Authors' response: Following the suggestion, in the revision, we added a sentence to reflect the reprocess of the apo data set in the C1 symmetry.

"Previously, we reported the structure of mBcs1L in the absence of nucleotide and substrate (Apo mBcs1L) determined by cryo-EM with imposed C7 symmetry (37). When we imposed no symmetry (C1) in reprocessing this data set (Supplementary Fig. 8) ..."

Reviewer 3's comment: - Extended Data Figure 7A contains a typo: the amount of particles for

ATP state-2 is 13,851, not 213,851.

Authors' response: corrected.

Reviewer 3's comment: -Extended data Fig. 10 shows a modeled PE lipid, but no fit is shown in the experimental map. I think its necessary to show the actual fit in the map as is done for ATP molecules in Figure 2.

Authors' response: We add a figure (Supplementary Figure 10B) to show the density and fitting.

Reviewer 3's comment: - The discussion would benefit from a brief mention of the possibility of stochastic rather than sequential ATP hydrolysis for some AAA ATPase family members (see second comment on Introduction).

Authors' response: we added a discussion in the revision as follows:

"Substrate translocation is thought to go through sequential ATP hydrolysis, as individual subunit transitions from ATP-bound form to extended ADP-bound form, thus moving downwards while pulling the substrate along stepwise. However, this sequential mechanism faces difficulties with the observations from some AAA proteins like RavA and ClpX, which feature two separate seam subunits in cryo-EM structures, leading to the proposed probabilistic mechanism for ATP hydrolysis (16-21)."

Reviewer 3's comment: - On p.22, the sentence "Recent and current structural studies have offered important insights into how Bcs1 meets these requirements and are in support of the concerted translocation mechanism is missing references.

Authors' response: Reference added. (Kater et al., Tang et al., 2020, NSMB; Pan et al., Nat Comm, 2023)

Reviewer 3's comment- For all of the processed cryoEM datasets and structural refinements, map-to-model FSC curves are absent. Map-to-model FSC curves should be calculated using unsharpened and unfiltered maps, and the FSC0.5 values should be reported in table 1 for each dataset.

- In Table 1 it is not stated which B-factors were used for sharpening of the maps.

Authors' response: As suggested, map-to-model FSC curves calculated using unmasked, unsharpened, and unfiltered maps have been added to supplementary figures 4, 5, 7, and 8, and

the FSC0.5 values were also reported in Table 1. We also added in Table 1 the B factors that were used to sharpen the maps.

Reviewer 3's comment - FSC curves shown in the workflow figures do not state what mask is used to calculate the FSC curves. For the refinements performed in cryoSPARC I assume the shown FSC curves are FSC curves calculated using a tight mask with correction by noise substitution? This should be explicitly mentioned.

Authors' response: Yes. For the refinements performed in cryoSPARC, FSC curves were calculated using a tight mask with correction by noise substitution. For the refinements performed in cisTEM, the FSC curves are calculated using a generous spherical mask with subsequent solvent correction inside the mask [Sindelar and Grigorieff, 2012; Grant et al. 2018]. The information is added to the Materials and Methods in the revision with indicated references.

Reviewer 3's comment – Line 641: 'Ab Initial model generation' should be replaced by 'Ab Initio model generation'.

Authors' response: corrected.

Reviewer 3's comment – Line 645: 'The best class of the ATP state particles were refined' should be changed to 'The best class of the ATP state particles was refined'

Authors' response: corrected.

Reviewer 3's comment – Line 649: 'The rest datasets' should be changed to 'The rest of the datasets' or 'The remaining datasets'.

Authors' response: corrected.

REVIEWER COMMENTS

Reviewer #1 (Remarks to the Author):

Reviewer #1

The authors have made cosmetic revisions to the manuscript text on the basis of my and the other reviewers' comments.

The clustering of the mutations causing the severest CIII deficiency phenotypes at the border between the TM region and Bcs1-specific region and at the subunit interface (Supplementary Fig. 11) is interesting.

It is true that many BCS1L patient mutations are compound heterozygous. However, there are at least four known homozygous missense mutations (M48V, S78G, P99L, S277N) that could be studied structurally. Additionally, those compound heterozygous cases, in which the other mutation is missense/truncating, likely result in a BCS1L heptamer only containing the point mutant protein. See Hikmat et al. 2021.

Of note, the S78G mutant protein is lost or decreased in both patient (Kotarsky et al. 2010) and mouse (Leveen et al. 2011) liver, suggesting that it is unstable due to misfolding or misassembly, and that such fully mutant heptamer may therefore never occur in vivo.

The paper is interesting but still lacks sufficient novelty compared to the groups's past work and interest to a broader readership.

Hikmat O, Isohanni P, Keshavan N, Ferla MP, Fassone E, Abbott MA, Bellusci M, Darin N, Dimmock D, Ghezzi D, Houlden H, Invernizzi F, Kamarus Jaman NB, Kurian MA, Morava E, Naess K, Ortigoza-Escobar JD, Parikh S, Pennisi A, Barcia G, Tylleskär KB, Brackman D, Wortmann SB, Taylor JC, Bindoff LA, Fellman V, Rahman S. Expanding the phenotypic spectrum of BCS1L-related mitochondrial disease. *Ann Clin Transl Neurol.* 2021 Nov;8(11):2155-2165.

Kotarsky H, Karikoski R, Mörgelin M, Marjavaara S, Bergman P, Zhang DL, Smet J, van Coster R, Fellman V. Characterization of complex III deficiency and liver dysfunction in GRACILE syndrome caused by a BCS1L mutation. *Mitochondrion.* 2010 Aug;10(5):497-509. doi: 10.1016/j.mito.2010.05.009.

Leveen P, Kotarsky H, Mörgelin M, Karikoski R, Elmér E, Fellman V. The GRACILE mutation introduced into Bcs1l causes postnatal complex III deficiency: a viable mouse model for mitochondrial hepatopathy. *Hepatology.* 2011 Feb;53(2):437-47.

Reviewer #2 (Remarks to the Author):

Response to revisions:

I appreciate the authors' rebuttal and clarifications, as well as their responses to reviewers 1 and 3. However, this does not significantly alleviate my concerns about the overall rather limited advance of this study and the in part insufficiently supported conclusions.

The presented structures primarily confirm the previously described concerted mechanism for conformational transitions of the Bcs1 heptamer. Proving a truly concerted ATP hydrolysis mechanism in all 7 subunits is more difficult. It is impossible to identify ADP bound to 1 or 2 out of 7 subunits when applying C7 symmetry for higher resolution, but even for the non-symmetrized reconstruction (C1) the presence of ADP in a small subset of subunits cannot be ruled out. The high conformational symmetry of the mBcs1L heptamer in the presence of ATP makes it impossible to properly align particles based on small differences like ADP versus ATP in some of the subunits, such that the nucleotide densities get averaged out. It is certainly conceivable that ATP hydrolysis in the mBcs1L heptamer indeed occurs in a truly concerted manner, but, nevertheless, the authors should somewhat qualify their statements and consider the difficulties of detecting different nucleotides at low abundance in a conformationally symmetric ring.

The authors' statement that their "EM work used a time-resolved approach to trap reaction intermediates" is quite an overstatement. Freezing mBcs1L at a single time point, 10 seconds after the addition of ATP, simply provides a steady-state hydrolysis snapshot of the ATPase and is unlikely to trap reaction intermediates. In fact, it is unclear why the authors were particular about taking a very early time point and concerned about a potential ADP accumulation later on, as they used only 86 nM mBcs1L heptamer in their grid preparation. This concentration of mBcs1L hydrolyzes just 5 μM ATP per minute, which is 0.25 % of the provided 2 mM ATP and most likely much lower than the typical concentration of contaminating ADP in an ATP solution. Also, the authors report an ATP hydrolysis rate of 40-60 ATP per heptamer per minute, but it is unclear at what temperature this activity was measured. If this refers to room temperature or 30 deg C, then ATP hydrolysis on ice will be even slower and ADP accumulation is even less of a concern.

A new finding of this paper is the visualization of the IMS seal structure in the transmembrane region. However, as mentioned in my original review, one has to be careful with interpreting conformational differences or helix angles in the absence of a membrane. For instance, in the ATP state-2 reconstruction, the N-terminal 3-turn helix apparently adopts a conformation that would make it enter the membrane (Fig. 5), which seems unlikely. Furthermore, the transmembrane region and IMS seal were completely unresolved in the reconstruction of ATP state-1, and this disorder was interpreted as functionally relevant without any experimental evidence.

There is no support for the proposed model that ATP state 1 represents a transitional state between ATP state 2 and the apo/ADP state. Both presented ATP states 1 and 2 are occupied with ATP and are substrate-free, such that it remains unclear what actually indicates state 2 to be the post-translocation state and what would drive the transitions from state 1 to 2. In the absence of further experimental

evidence, e.g. intermediates with bound substrate, it is unclear whether states 1 and 2 even represent actual translocation states, and it is pure speculation to propose a particular order of these states during ISP precursor translocation, solely based on the conformations of beta sheets 1 and 2, or the observation of an apparent opening versus partial opening of the matrix and IMS seals.

Minor: The authors propose a substrate-release mechanism that includes mBcs1L subunit dissociation. In this context, they reference canonical AAA proteins such as p97 “that release the ‘seam subunit’ after ATP hydrolysis”. This is misleading, as neither p97 nor other canonical AAA proteins have been shown to “release” subunits during the ATPase cycle. Post-hydrolysis subunits are assumed to vertically move in a spiral staircase arrangement, which makes their interfaces to neighboring subunits appear less defined. Although this may open up a transient inter-subunit seam for lateral substrate exit, there is no indication for subunits being released from the ring, and the authors should clarify this.

In summary, I leave it to the editor to decide whether the presented results, in particular the newly described IMS seal structure and the low-resolution ISP-ED substrate density observed for ADP-bound mBcs1L, represent an advance sufficient for a publication in Nature Communication.

Reviewer #3 (Remarks to the Author):

The Authors have sufficiently addressed my questions, suggestions and technical comments. I recommend publication of the revised manuscript.

With best regards,

Jan Felix

REVIEWER COMMENTS

Reviewer #1

Comment from Reviewer 1: The authors have made cosmetic revisions to the manuscript text on the basis of my and the other reviewers' comments. The clustering of the mutations causing the severest CIII deficiency phenotypes at the border between the TM region and Bcs1-specific region and at the subunit interface (Supplementary Fig. 11) is interesting.

It is true that many BCS1L patient mutations are compound heterozygous. However, there are at least four known homozygous missense mutations (M48V, S78G, P99L, S277N) that could be studied structurally. Additionally, those compound heterozygous cases, in which the other mutation is missense/truncating, likely result in a BCS1L heptamer only containing the point mutant protein. See Hikmat et al. 2021.

Of note, the S78G mutant protein is lost or decreased in both patient (Kotarsky et al. 2010) and mouse (Leveen et al. 2011#681) liver, suggesting that it is unstable due to misfolding or misassembly, and that such fully mutant heptamer may therefore never occur in vivo.

The paper is interesting but still lacks sufficient novelty compared to the groups's past work and interest to a broader readership.

Hikmat O, Isohanni P, Keshavan N, Ferla MP, Fassone E, Abbott MA, Bellusci M, Darin N, Dimmock D, Ghezzi D, Houlden H, Invernizzi F, Kamarus Jaman NB, Kurian MA, Morava E, Naess K, Ortigoza-Escobar JD, Parikh S, Pennisi A, Barcia G, Tylleskär KB, Brackman D, Wortmann SB, Taylor JC, Bindoff LA, Fellman V, Rahman S. Expanding the phenotypic spectrum of BCS1L-related mitochondrial disease. *Ann Clin Transl Neurol.* 2021 Nov;8(11):2155-2165.

Kotarsky H, Karikoski R, Mörgelin M, Marjavaara S, Bergman P, Zhang DL, Smet J, van Coster R, Fellman V. Characterization of complex III deficiency and liver dysfunction in GRACILE syndrome caused by a BCS1L mutation. *Mitochondrion.* 2010 Aug;10(5):497-509. doi: 10.1016/j.mito.2010.05.009.

Leveen P, Kotarsky H, Mörgelin M, Karikoski R, Elmér E, Fellman V. The GRACILE mutation introduced into Bcs1l causes postnatal complex III deficiency: a viable mouse model for mitochondrial hepatopathy. *Hepatology.* 2011 Feb;53(2):437-47.

Authors' Response: We appreciate Reviewer1's comment and the 3 papers describing clinical studies of disease-causing mutations in Bcs1L in the literature. In particular, the work by Hikmat et al 2021 emphasized the observation of the S78G mutation in the early onset of Bcs1L-associated diseases and brought us up to date of more recently identified Bcs1 mutants. It is this kind of clinical studies that are highly motivating to us in our investigation of structural and biochemical mechanisms underlying the changes in function by mutations of Bcs1L. Our previous

studies on the human AAA protein p97 harboring IBMPFD mutations are good examples (Tang et al., 2010 EMBO, and Tang et al., 2017, Cell Discovery).

We agree with the reviewer that some of the reported disease-associated mutations are indeed homozygous and are amenable for structural studies. As mentioned in our previous response, we have made *Pichia* expression constructs of several mutant Bcs1L, including S78G, R155P, R183H, and R183C, in which S78G and R183H were selected precisely for their being homozygous. We noted the reviewer's comment on the low detection level of the S78G mutant in patient and in mouse model. Our purification experiments suggest that this mutant was expressed at a similar level to the wild type Bcs1L. However, purified mutant protein is prone to aggregation and was more difficult for structural characterization. Thus, the study of these mutants is still in its early stage, and we think it is inappropriate to have premature results included in this paper.

Given the additional disease-causing mutations identified in the paper by Hikmat et al. 2021 and the new structural information of Bcs1L obtained in this work, we revised the Figure S11 by mapping all mutations known to us onto the complete Bcs1L structure in the ADP conformation. In this figure, we added mutations N13S, located in the IMS seal, and R33Q, located in the middle of the TM helix. Both mutations cause the severe form of mitochondrial diseases and were not illustrated in our previous publication due to structural disorders in these two regions (Tang et al., 2020). Other new mutations were also added to the Bcs1-specific domain or to the AAA domain. With the available information about new mutations and associated clinical symptoms, it becomes clear that mutations in the Bcs1 specific domain, especially those close to the junction between TMH and Bcs1 specific domain, are more likely to predispose patients to the severe form of the Bcs1L-linked mitochondrial disorders. We also revised the legend to the Figure S11 accordingly.

We also added the reference 36 to the Introduction section and reference 54 to the Discussion section.

36. Fellman, V., Rapola, J., Pihko, H., Varilo, T., and Raivio, K. O. (1998) Iron-overload disease in infants involving fetal growth retardation, lactic acidosis, liver haemosiderosis, and aminoaciduria Lancet 351, 490-493 10.1016/S0140-6736(97)09272-6

54. Hikmat, O., Isohanni, P., Keshavan, N., Ferla, M. P., Fassone, E., Abbott, M. A. et al. (2021) Expanding the phenotypic spectrum of BCS1L-related mitochondrial disease Ann Clin Transl Neurol 8, 2155-2165 10.1002/acn3.51470

The following sentence in the Discussion section is also revised:

"The need to maintain the membrane integrity is underscored by the observation that all eight mutations (R45C, M48V, T50A, R73C, S78G, P99L, R109W, and R144Q) identified from patients with GRACILE syndrome (38, 54), a severe form of Bcs1L-linked mitochondrial disease, are

clustered at the border between the TM region and Bcs1-specific region and at the subunit interface (Fig. S11)."

Reviewer #2 (Remarks to the Author):

Comment from Reviewer 2: I appreciate the authors' rebuttal and clarifications, as well as their responses to reviewers 1 and 3. However, this does not significantly alleviate my concerns about the overall rather limited advance of this study and the in part insufficiently supported conclusions.

The presented structures primarily confirm the previously described concerted mechanism for conformational transitions of the Bcs1 heptamer. Proving a truly concerted ATP hydrolysis mechanism in all 7 subunits is more difficult. It is impossible to identify ADP bound to 1 or 2 out of 7 subunits when applying C7 symmetry for higher resolution, but even for the non-symmetrized reconstruction (C1) the presence of ADP in a small subset of subunits cannot be ruled out. The high conformational symmetry of the mBcs1L heptamer in the presence of ATP makes it impossible to properly align particles based on small differences like ADP versus ATP in some of the subunits, such that the nucleotide densities get averaged out. It is certainly conceivable that ATP hydrolysis in the mBcs1L heptamer indeed occurs in a truly concerted manner, but, nevertheless, the authors should somewhat qualify their statements and consider the difficulties of detecting different nucleotides at low abundance in a conformationally symmetric ring.

Authors' response: The reviewer expressed a doubt about ATP molecules identified in our structure. During EM data processing, we either imposed the 7-fold symmetry (C7) or no symmetry (C1), resulting EM densities that are compatible with ATP (Figure 2B). As shown in Table 1, the modeled ATP molecules have lower average B factors than protein atoms for both C7 and C1 structures, indicating these bound ATP molecules are well ordered. Consistently, the ATP models gave higher average CC values 0.93 and 0.92, respectively, for C1 and C7 structures, which are much better for proteins. These are the commonly used indicators for quality control of structural models, and they are included in the Table 1 for a reason. Furthermore, as the reviewer has acknowledged, we and others have established an interdependence between nucleotide state and Bcs1 conformation in previous publications (Tang et al., 2020, NSMB, Katie et al., 2020, NSMB, Pan et al., 2023, Nat Comm). Such interdependences between the type of bound nucleotide and corresponding protein conformation have been well established for many AAA proteins. In our present work, the identity of bound nucleotide was established based on not only the EM density in both C1 and C7 forms, but also the conformations of Bcs1L, which are dramatically different between the ADP- and ATP-bound forms. These nucleotide dependent conformations have been confirmed by X-ray diffraction as well as by cryo-EM for both full-length Bcs1 and truncated AAA domains at high resolutions (up to 2.2 Å, Tang et al., 2020, NSMB).

Comment from Reviewer 2: The authors' statement that their "EM work used a time-resolved approach to trap reaction intermediates" is quite an overstatement. Freezing mBcs1L at a single time point, 10 seconds after the addition of ATP, simply provides a

steady-state hydrolysis snapshot of the ATPase and is unlikely to trap reaction intermediates. In fact, it is unclear why the authors were particular about taking a very early time point and concerned about a potential ADP accumulation later on, as they used only 86 nM mBcs1L heptamer in their grid preparation. This concentration of mBcs1L hydrolyzes just 5 μM ATP per minute, which is 0.25 % of the provided 2 mM ATP and most likely much lower than the typical concentration of contaminating ADP in an ATP solution. Also, the authors report an ATP hydrolysis rate of 40-60 ATP per heptamer per minute, but it is unclear at what temperature this activity was measured. If this refers to room temperature or 30 deg C, then ATP hydrolysis on ice will be even slower and ADP accumulation is even less of a concern.

Authors' response: A major goal of this work is to experimentally validate if ATP hydrolysis by Bcs1 follows the sequential/random mechanism as other canonical AAA-ATPases. Our approach is to capture different conformations while Bcs1 is undergoing active ATP hydrolysis. We hope the reviewer is convinced that Bcs1L is a slow ATPase with an average lifetime for the ATP state of approximately 5.6 s (Pan et al., 2023, Nat Comm), which is similar to many other well-studied AAA proteins. We also concur with the reviewer that the ATPase reaction carried out by Bcs1L, like most biochemical reactions, has likely reached a steady-state prior to being applied to the grid and frozen. By freezing grids at early and later stages of ATPase reaction, we hope to capture any additional conformations other than the uniform ADP- and ATP-bound states, which would make a distinction between the sequential/random mechanism and concerted mechanism (explained in Figure 2A). In previously published work, canonical AAA proteins were captured in non-planar conformations using similar approaches, which led to the proposal of sequential/random mechanism.

To specifically address reviewer's question on the difference between the AFM approach and the current work, we used the term "time-resolved" to distinguish our approach from the AFM method that allowed real-time visualization. In fact, in our manuscript, we refrained from using the term "time-resolved". As to why we used 10 second incubation, previous report (Kater et al., 2020, NSMB) and our current work showed that extended incubation time with ATP led to the dominance of ADP conformation, which we suggested in our previous response to the reviewer 3's comment to be likely a consequence of product (ADP) inhibition (slow release).

Comment from Reviewer 2: A new finding of this paper is the visualization of the IMS seal structure in the transmembrane region. However, as mentioned in my original review, one has to be careful with interpreting conformational differences or helix angles in the absence of a membrane. For instance, in the ATP state-2 reconstruction, the N-terminal 3-turn helix apparently adopts a conformation that would make it enter the membrane (Fig. 5), which seems unlikely. Furthermore, the transmembrane region and IMS seal were completely unresolved in the reconstruction of ATP state-1, and this disorder was interpreted as functionally relevant without any experimental evidence.

Authors' response: We understand the reviewer's point that our measurement of TM helix movement from the ADP state to ATP state-2 may not be functionally relevant since our

structures were determined in detergent micelles rather than in lipid bilayer. Although we have worked on many membrane protein structures and have yet to see an example in our detergent-derived structures that grossly violates structural principles, examples of altered subunit associations could be found in the literature between detergent- and lipid-embedded structures such as the structures of bacteriorhodopsin and small multidrug resistant membrane protein EmrE. Unlike Bcs1L, these oligomeric small membrane proteins often do not have large extramembrane domains, thus leaving their subunit association state vulnerable to detergent treatment. By contrast, membrane proteins with very large extramembrane domains normally show insignificant detergent-induced changes.

For Bcs1L, if the reviewer is familiar with our AFM work (Pan et al., 2023, Nat Comm), the isolated Bcs1L was reconstituted into a membrane bilayer before measurements were made. It was observed that the IMS side of the heptamer underwent cycles of changes in protrusion height and diameter in the presence of ATP, indicating that there might be a membrane deformation or membrane-thinning event surrounding the TM region of Bcs1L when it transitions from ATP to ADP or vice versa. The changes in height and diameter of the IMS seal is consistent with our observation of TM helices tilting while Bcs1L is transitioning from one nucleotide type to another.

In the revision, we revised the following sentence in the penultimate paragraph in the Result section: "This difference in ϕ results in a shallower TM basket and a slightly more opened IMS-seal with a diameter of approximately 13 Å (Fig.5B and 5C)" to "This difference in ϕ results in a shallower TM basket and a slightly more opened IMS-seal with a diameter of approximately 13 Å (Fig.5B and 5C), which is consistent with a previous observation by AFM showing the IMS seal undergoing an up-and-down movement during turnover (41)."

Comment from Reviewer 2: There is no support for the proposed model that ATP state 1 represents a transitional state between ATP state 2 and the apo/ADP state. Both presented ATP states 1 and 2 are occupied with ATP and are substrate-free, such that it remains unclear what actually indicates state 2 to be the post-translocation state and what would drive the transitions from state 1 to 2. In the absence of further experimental evidence, e.g. intermediates with bound substrate, it is unclear whether states 1 and 2 even represent actual translocation states, and it is pure speculation to propose a particular order of these states during ISP precursor translocation, solely based on the conformations of beta sheets 1 and 2, or the observation of an apparent opening versus partial opening of the matrix and IMS seals.

Authors' response: In our previous report (Tang et al., 2020, NSMB), we proposed the following:

In this proposed mechanism, substrate ISP binds to apo form of Bcs1 because the size of the matrix cavity in the apo conformation is sufficiently large to accommodate the ISP-ED. ATP γ S binding leads to a dramatic conformational change that reduces the size of the matrix cavity by 2/3. This ATP γ S bound structure is equivalent to that of the ATP state-1 conformation in this work, featuring disordered TM helices and IMS seal. ATP hydrolysis restores the apo conformation. Based on this proposed model, we add ATP state-2 conformation between ATP state-1 and ADP conformation, because ATP state-2 has a better ordered TM domain and IMS seal.

Comment from Reviewer 2: Minor: The authors propose a substrate-release mechanism that includes mBcs1L subunit dissociation. In this context, they reference canonical AAA proteins such as p97 “that release the ‘seam subunit’ after ATP hydrolysis”. This is misleading, as neither p97 nor other canonical AAA proteins have been shown to “release” subunits during the ATPase cycle. Post-hydrolysis subunits are assumed to vertically move in a spiral staircase arrangement, which makes their interfaces to neighboring subunits appear less defined. Although this may open up a transient inter-subunit seam for lateral substrate exit, there is no indication for subunits being released from the ring, and the authors should clarify this.

Authors' response: We meant to say that the D2 domain of the "seam" subunit dissociate. It is corrected.

In summary, I leave it to the editor to decide whether the presented results, in particular the newly described IMS seal structure and the low-resolution ISP-ED substrate density observed for ADP-bound mBcs1L, represent an advance sufficient for a publication in Nature Communication.

Reviewer #3 (Remarks to the Author):

Comment from Reviewer 3: The Authors have sufficiently addressed my questions,

suggestions and technical comments. I recommend publication of the revised manuscript.

Authors' response: We thank the Reviewer 3 for the support.

REVIEWERS' COMMENTS

Reviewer #2 (Remarks to the Author):

Although I appreciate the authors' rebuttal and responses, they did not alleviate my concerns raised in the last review or the general concern about the somewhat limited advance of this study. As the work appears solid from a technical perspective, I will have to leave it to the editor to decide whether the limited novelty and advance is sufficient for publication in Nature Communications.

REVIEWER COMMENTS

Reviewer #2

Comment from Reviewer 2: Although I appreciate the authors' rebuttal and responses, they did not alleviate my concerns raised in the last review or the general concern about the somewhat limited advance of this study. As the work appears solid from a technical perspective, I will have to leave it to the editor to decide whether the limited novelty and advance is sufficient for publication in Nature Communications.

Previous Comment from Reviewer 2: There is no support for the proposed model that ATP state 1 represents a transitional state between ATP state 2 and the apo/ADP state. Both presented ATP states 1 and 2 are occupied with ATP and are substrate-free, such that it remains unclear what actually indicates state 2 to be the post-translocation state and what would drive the transitions from state 1 to 2. In the absence of further experimental evidence, e.g. intermediates with bound substrate, it is unclear whether states 1 and 2 even represent actual translocation states, and it is pure speculation to propose a particular order of these states during ISP precursor translocation, solely based on the conformations of beta sheets 1 and 2, or the observation of an apparent opening versus partial opening of the matrix and IMS seals.

Authors' Response: We appreciate Reviewer2's comments and understand his/her concerns over the lack of additional supporting evidence for the functional relevance of the two observed conformations in ATP-bound states. In this revision, we changed the last sentence in the Results section from: "The conformation of β -Sheet II in ATP state-1 is partially extended with the density at the tip being less well-defined, which may represent a transitional state between ATP-2 and apo/ADP state (Fig. 5A, top)." to "The conformation of β -Sheet II in ATP state-1 is partially extended with the density at the tip being less well-defined (Fig. 5A, top). The functional relevance of these two observed ATP states remains to be seen."

We also revised the last paragraph in the Discussion section from "(2) ATP binding to all seven subunits initiates a concerted conformational change in Bcs1L, opening both Matrix-seal and IMS-seal and pushing the ISP-ED across the membrane, as represented by the ATP state-1. (3) Conceivably, ATP state-1 transits quickly to ATP state-2, the post-translocation state, where Bcs1 captures ISP precursor by its TM helix." to "(2) ATP binding to all seven subunits initiates a concerted conformational change in Bcs1L, opening both Matrix-seal and IMS-seal and pushing the ISP-ED across the membrane, as represented by the ATP state-1 or ATP state-2. (3) Conceivably, in ATP-bound state, the post-translocation state, Bcs1 captures ISP precursor by its TM helix."